# *Egr-5* is a post-mitotic regulator of planarian epidermal differentiation

**Kimberly C Tu[1], Li-Chun Cheng[1], Hanh TK Vu[1], Jeffrey J Lange[1], Sean A McKinney[1], Chris W Seidel[1], Alejandro Sánchez Alvarado[1,2]***

[1]Stowers Institute for Medical Research, Kansas City, United States; [2]Howard Hughes Medical Institute, Stowers Institute for Medical Research, Kansas City, United States

**Abstract** Neoblasts are an abundant, heterogeneous population of adult stem cells (ASCs) that facilitate the maintenance of planarian tissues and organs, providing a powerful system to study ASC self-renewal and differentiation dynamics. It is unknown how the collective output of neoblasts transit through differentiation pathways to produce specific cell types. The planarian epidermis is a simple tissue that undergoes rapid turnover. We found that as epidermal progeny differentiate, they progress through multiple spatiotemporal transition states with distinct gene expression profiles. We also identified a conserved early growth response family transcription factor, *egr-5*, that is essential for epidermal differentiation. Disruption of epidermal integrity by *egr-5* RNAi triggers a global stress response that induces the proliferation of neoblasts and the concomitant expansion of not only epidermal, but also multiple progenitor cell populations. Our results further establish the planarian epidermis as a novel paradigm to uncover the molecular mechanisms regulating ASC specification in vivo.

## Introduction

Adult stem cells (ASCs) are tissue-specific cells with the capacity to self-renew and differentiate to continually replace cells lost to normal physiological turnover or injury. As a result, ASCs play an essential role in preserving the anatomical form and function of most multicellular organisms. The precise coordination of stem cell proliferation and proper fate specification is of paramount importance to tissue growth and organismal homeostasis. Excessive stem cell divisions can lead to tumorigenesis (*Visvader and Lindeman, 2012*), while a loss in proliferation capacity can contribute to premature aging (*Gopinath and Rando, 2008*). Understanding the cellular and molecular mechanisms that regulate the balance between stem cell proliferation, differentiation, and cell death will thus provide fundamental insights into tissue maintenance and repair. It will also illuminate the molecular basis of tissue dysfunction, including disease progression and aging.

The model planarian *Schmidtea mediterranea* has emerged as an experimental system that provides a unique window into major aspects of *in vivo* stem cell biology, including regeneration, fate determination and homeostatic plasticity (*Rink, 2013*; *Roberts-Galbraith and Newmark, 2015*). Neoblasts, the planarian stem cells, are in a state of perpetual action. They are widely distributed throughout the body mesenchyme, driving constitutive renewal of tissues during homeostasis and endowing planarians with the remarkable capacity to regenerate wholly from tiny tissue fragments (*Brøndsted, 1969*; *Newmark and Sánchez Alvarado, 2000*; *Wagner et al., 2011*). Neoblasts, the only dividing cells in planarians, are believed to be collectively comprised of both a heterogeneous population of pluripotent cells with broad differentiation potential and also lineage-committed progenitor cells that give rise to specific tissues (*Hayashi et al., 2010*; *Scimone et al., 2014*; *van Wolfswinkel et al., 2014*; *Wagner et al., 2011*). To ensure the integrity of adult tissues during

*For correspondence: asa@stowers.org

**eLife digest** Tissues in adult animals contain cells called adult stem cells, which can divide to generate more adult stem cells (in a process called self-renewal) or specialize into other cell types (via a process called differentiation). This means that adult stem cells can replace the specialized cells that are continually lost from animal tissues and organs. This allows the organs to continue to work properly. It is important to understand how adult stem cells decide whether to self-renew or differentiate because if they proliferate too much they may form abnormal growths such as tumors. On the other hand, if adult stem cells do not properly differentiate into specialized cells it can lead to tissue degeneration or even premature aging.

Now Tu et al. have used planarian flatworms, which are considered masters of regeneration, as a model to study how adult stem cells differentiate into more specialized cells. In particular, the experiments explored how the flatworm's adult stem cells (which are called neoblasts) develop into the epidermal cells that form the equivalent of the worm's skin.

Tu et al. show that when a neoblast becomes a mature epidermal cell, it has to undergo multiple transition steps. Slightly different genes are expressed during each step, but a gene called *egr-5* controls the expression of all of these marker genes. The *egr-5* gene is highly expressed when cells start to develop into epidermal cells. Reducing this gene's activity blocks the cells from differentiating properly, meaning that they do not form mature epidermal cells. The loss of new epidermal cells causes a disruption in the overall integrity of the worm's outer surface and this triggers a wound response throughout the whole animal. The neoblasts in turn respond by proliferating excessively and generating other differentiated cells such as neurons and gut cells. However, without *egr-5*, the flatworms still cannot make new epidermal cells and they ultimately die.

The findings highlight that the development of epithelial cells in this relatively simple organism is much more complicated than suspected. In the future, it will be important to understand how the *egr-5* gene controls the proper differentiation and maturation of epidermal cells and whether these mechanisms are conserved in other animals.

homeostasis and regeneration, neoblasts must perpetuate themselves and generate lineage-committed progenitor cells that give rise to precise numbers of differentiated cell types in a proper spatial and temporal sequence.

A general principle used to establish planarian lineages has been to identify tissue-specific transcription factors (TF) expressed in subsets of neoblasts (*smedwi-1*[+]) (*Reddien et al., 2005b*) that are also required for the specification of those tissues, including the eye (*Lapan and Reddien, 2011*; *Lapan and Reddien, 2012*), protonephridia (*Scimone et al., 2011*), pharynx (*Adler et al., 2014*; *Scimone et al., 2014*), and discrete neuronal sub-types (*Cowles et al., 2013*; *Currie and Pearson, 2013*; *Marz et al., 2013*; *Scimone et al., 2014*; *Wenemoser et al., 2012*). These TFs have typically been identified through evolutionary conservation, induced expression in neoblasts during regeneration, or through transcriptional profiling of isolated tissues. Additional methods including BrdU incorporation (*Newmark and Sánchez Alvarado, 2000*), perdurance of the SMEDWI-1 protein in differentiating progeny cells (*Guo et al., 2006*; *Wenemoser and Reddien, 2010*; *Zhu et al., 2015*), and gamma irradiation (*Eisenhoffer et al., 2008*) have also been used to link neoblasts with their progeny. Although RNAi knockdown of lineage-committed TFs blocks the regeneration of their specified tissues, many of these TFs are expressed throughout those tissues, making it difficult to study how different cell types within the same tissue are formed. Therefore, elucidating the cellular mechanisms that bridge the pluripotent and the differentiated state remains a challenge.

The planarian epidermis is a simple, monostratified tissue that consists of clear histological organization of multiple differentiated multi-ciliated and non-ciliated cell types (*Rompolas et al., 2010*). Individual epidermal cells must continuously be replaced due to damage from environmental insults and exogenous wounds. To replenish these cells, neoblasts residing deep in the mesenchyme must produce cells that mobilize, undergo multiple determination steps, cross the basement membrane, intercalate and differentiate into the epidermis. Moreover, the epidermis provides the critical first step in regeneration by covering the amputation-induced wound site through cell spreading (*Morita and Best, 1974*). Recent work has identified a prominent class of neoblasts referred to as

zeta-class possessing a distinct molecular signature, including the novel zinc-finger gene *zfp-1* (*van Wolfswinkel et al., 2014*). *Zfp-1*(RNAi) animals can regenerate tissues including the gut, brain and muscle, but fail to generate cells expressing epidermal markers, indicating that zeta-class neoblasts likely give rise to an epidermal lineage.

*Zfp-1*, along with the chromatin remodeling factor *chd4*, the tumor suppressor gene *p53*, and most recently an RNA-binding protein *mex3-1*, have all been shown to be required for the maintenance of two related postmitotic, sub-epidermal cell populations expressing the specific marker genes *prog-1* and *AGAT-1* (*Pearson and Sánchez Alvarado, 2010*; *Scimone et al., 2010*; *Wagner et al., 2012*; *Zhu et al., 2015*). These abundant *prog-1*[+] and *AGAT-1*[+] cell populations, originally identified as early and late progeny cells based on their rapid turnover kinetics (*Eisenhoffer et al., 2008*), have been widely used to assess neoblast differentiation. Given that zeta-class neoblasts are required for the generation of *prog-1*[+], *AGAT-1*[+] cells and other markers of epidermal cell types, *prog-1* and *AGAT-1* likely mark two major populations of epidermal progeny cells. However, it remains unclear whether the diverse cell types in the planarian epidermis all share common or distinct lineage relationships with each other, and the mechanisms that control the progression of epidermal progenitors along distinct differentiation paths into mature cell types are completely unknown.

To understand the molecular mechanisms underlying neoblast differentiation and how they give rise to multiple cell types, we devised a strategy to identify critical factors enriched in the *AGAT-1*[+] cell population required for epidermal lineage progression. We performed RNA-seq analysis of *chd4* and *p53* RNAi animals and characterized additional markers enriched in *AGAT-1*[+] and related post-mitotic cells. We find that epidermal progeny cells form distinct mesenchymal populations and undergo differentiation in a spatially and temporally graded manner into the mature epidermis. We also describe a conserved transcription factor of the early growth response family, *egr-5*, that is expressed in post-mitotic progeny cells and is an essential regulator of post-mitotic epidermal fate specification. Taken together, our results further establish the planarian epidermis as a paradigm to study adult lineage specification *in vivo*, contributing to our knowledge of mechanisms required for the proper execution of stem cell fate decisions.

## Results

### Identification of regulators of epidermal post-mitotic fate transitions

The *prog-1*[+] (early progeny) and *AGAT-1*[+] (late progeny) postmitotic cell populations constituting the first neoblast lineage described in planarians (*Eisenhoffer et al., 2008*) have been widely used as an assay for neoblast differentiation (*Fraguas et al., 2011*; *Pearson and Sánchez Alvarado, 2010*; *Scimone et al., 2010*; *Wagner et al., 2012*). The zeta-class neoblasts are a major subclass of planarian neoblasts identified molecularly by a specific gene signature, mainly two TFs, *zfp-1* and *soxP-3* (*van Wolfswinkel et al., 2014*). Zeta neoblasts are generated from the collectively pluripotent sigma-class neoblasts, and have recently been shown to generate *prog-1*[+] and *AGAT-1*[+] cells as well as other populations spanning the epidermis (*van Wolfswinkel et al., 2014*), suggesting that they are part of an epidermal lineage (*Figure 1A*). This lineage is a useful model to dissect the choreography of neoblast progeny differentiation because it is an abundant cell population undergoing rapid turnover, and both *prog-1*[+] and *AGAT-1*[+] cells are molecularly and spatially distinct. *Prog-1*[+] (early progeny) cells likely represent a very transient cell population because they are lost about two days after a lethal dose of irradiation, whereas *AGAT-1*[+] (late progeny) cells are not completely lost until about seven days post-irradiation (*Eisenhoffer et al., 2008*). Therefore, the precise molecular relationships between these cell populations and the mechanisms that control the progression and maturation of the zeta-class epidermal lineage still must be resolved.

To further characterize the molecular transitions implicated in epidermal cell maturation, we compared whole worm expression profiles from two different RNAi knockdown conditions known to drastically reduce the population of *AGAT-1*[+] cells. Previous studies have shown that the chromatin-remodeling factor *chd4* and the tumor suppressor gene *p53*, both expressed in the neoblast compartment, are required for tissue homeostasis and regeneration (*Pearson and Sánchez Alvarado, 2010*; *Scimone et al., 2010*). Both *chd4* and *p53* are also expressed in *AGAT-1*[+] cells. *Chd4* is expressed broadly throughout the animal parenchyma as well as in the brain and ventral nerve cords,

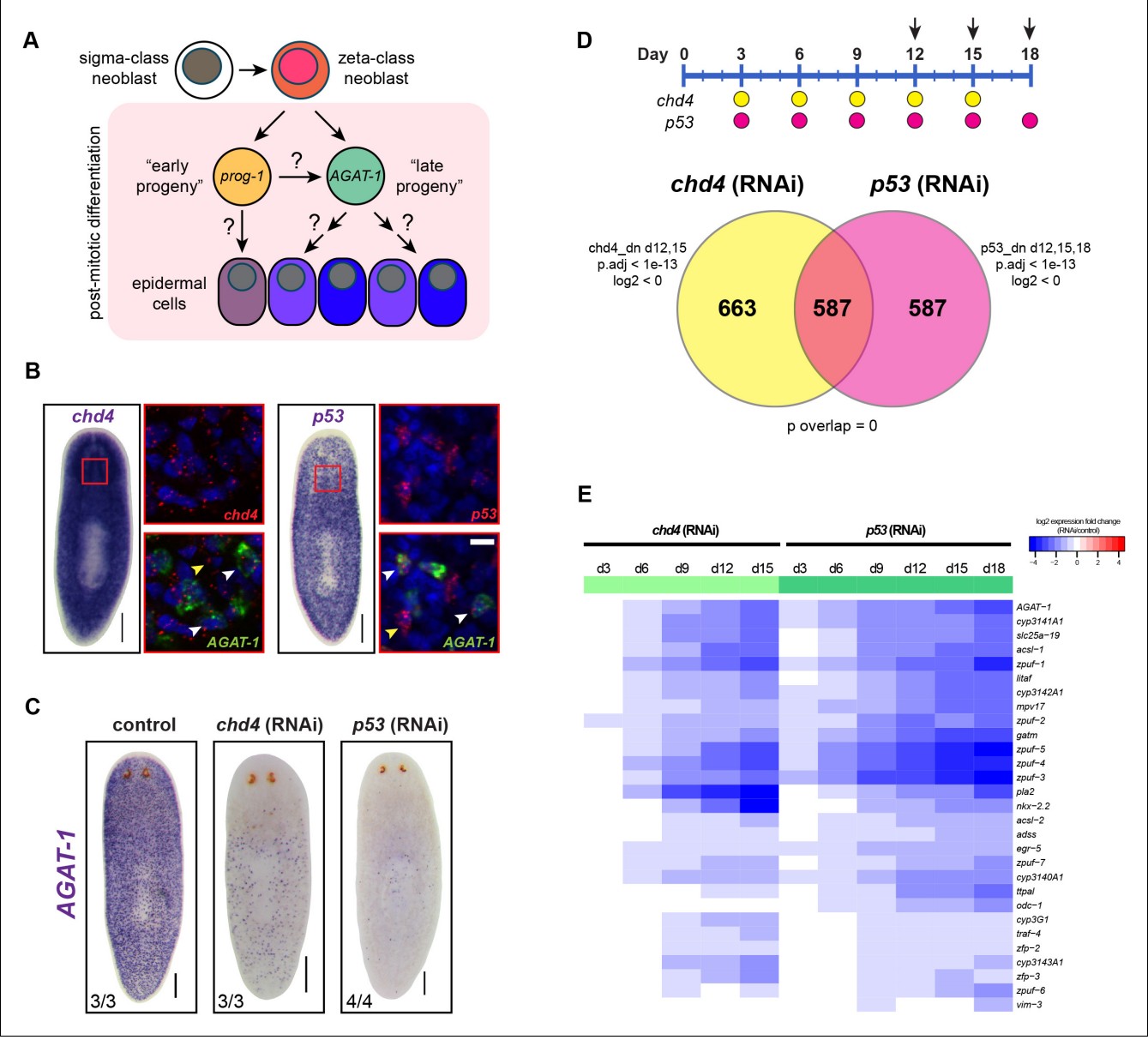

**Figure 1.** Identification of a common transcriptional down-regulated gene set in *chd4* and *p53* RNAi animals. (**A**) Current model of planarian epidermal lineage specification. Sigma-class neoblasts give rise to zeta-class neoblasts, which in turn generate *prog-1* (early progeny) and *AGAT-1* (late progeny) expressing cells. The precise molecular relationship between these cell types remains unclear, but collectively give rise to an unknown number of epidermal cell types through an unknown number of transitional states. (**B**) Whole-mount in situ (WISH) expression patterns of *chd4* and *p53* in wild-type planaria. Left panels: *chd4* colorimetric WISH and double fluorescent in situ (FISH) of *chd4* (red), *AGAT-1* (green) and DAPI (blue). Right panels: *p53* colorimetric WISH and double FISH of *p53* (red), *AGAT-1* (green) and DAPI (blue). Magnified regions are single confocal planes from boxed regions. White arrowheads highlight cells with co-localized expression of either *chd4* or *p53* and *AGAT-1*. Yellow arrowheads highlight additional mesenchymal cells that do not express *AGAT-1*. Scale bars: 200 µm; 10 µm (zoomed images). (**C**) *chd4*(RNAi) and *p53*(RNAi) result in the loss of *AGAT-1* expressing cells. Representative colorimetric WISH images shown at 3Fd18 of RNAi treatment. Scale bar: 200 µm. (**D**) Venn diagram of genes down-regulated in *chd4* and *p53* RNAi data sets. Timeline of RNAi treatment (Fed d0, d3, d6) and RNA collected for *chd4* (yellow circles) and *p53* RNAi (pink circles). Arrows highlight time points used to identify down-regulated gene set. Criteria used for genes to make the cut-off are shown. dn, down-regulated genes; p.adj, adjusted p-value; log2, fold change of RNAi over control (see Materials and methods). For the *chd4* and *p53* RNAi overlapping data set (587 genes), a hypergeometric distribution and a universe size of 28,668 was used to generate p-value for determining significance of overlap by chance. See also ***Supplementary file 1***. (**E**) Heat map depicting candidate genes that were selected from the *chd4*(RNAi) and *p53*(RNAi) down-regulated data sets for further characterization after in situ hybridization screen. log2 fold changes in RNAi expression relative to each control time point are shown.

*Figure 1. continued on next page*

*Figure 1. Continued*

The following figure supplements are available for Figure 1:

**Figure supplement 1.** RNA-seq validation of control genes in *chd4* and *p53* RNAi animals.

and *p53* is expressed in discrete mesenchymal cells that also include *smedwi-1⁺* and *prog-1⁺* cells (*Figure 1B*). RNAi knockdown of both *chd4* and *p53* results in a dramatic loss of *AGAT-1⁺* cells (*Figure 1C*). However, it is not well understood whether these mechanisms of action are direct or indirect. Based on their close proximity to the basement membrane and on post-irradiation kinetics, *AGAT-1⁺* cells likely represent a relatively stable transition at which multiple fate decisions could be made. Therefore, we reasoned that the union of *chd4* and *p53* RNAi whole-worm RNA-seq down-regulated datasets would contain genes enriched in *AGAT-1⁺* cells, as well as genes with enriched expression in epidermal cell populations that arise from *AGAT-1⁺* progeny cells, potentially revealing critical factors required for post-mitotic differentiation.

## Global expression analysis reveals a common *chd4* and *p53* RNAi down-regulated gene set

We compared whole worm gene expression profiles of *chd4* and *p53* RNAi animals to those of control animals through multiple time points of RNAi treatment (*Figure 1D*). To verify the specificity and sensitivity of the data sets, we examined a number of control genes (*smedwi-1, prog-1, AGAT-1* and *chd4, p53*) with known kinetics of disappearance after RNAi knockdown and found strong correlation (*Figure 1—figure supplement 1A,B*), indicating that known marker genes display predicted behaviors in our RNA-seq data.

To identify common down-regulated genes in *chd4* and *p53* RNAi-treated animals, we adopted criteria where such candidate genes would be required to have significant adjusted p-values (p.adj <1e-13) weighted towards multiple later time points after RNAi treatment (for *chd4*: d12, d15; for *p53*: d12, d15, d18). Using these parameters, a total of 1,250 genes were designated to be down-regulated after *chd4* RNAi treatment, 1,174 genes were down-regulated after *p53* RNAi treatment, and a total of 587 common genes were found to be significantly down-regulated in both data sets (*Figure 1D—figure supplement 1C,D*). Of the 587 common down-regulated genes, 70% (411/587) are predicted to encode homologs of proteins found in other organisms (*Supplementary file 1,2* for up-regulated genes). Earlier studies have reported genes involved in creatine metabolism (*AGAT-1, AGAT-2, AGAT-3*), polyamine biosynthesis (*odc-1*), and monooxygenase activities (*cyp1a1*), as well as others with unknown function, to be expressed in similar cell populations (*Eisenhoffer et al., 2008*; *van Wolfswinkel et al., 2014*; *Zhu et al., 2015*). We analyzed the common gene set by assigning gene ontology (GO) terms and found that small molecule metabolic processes and transporter activity were the most overrepresented biological processes and molecular functions (*Supplementary file 1*).

We selected approximately 150 genes from the common down-regulated data set to screen by whole-mount in situ hybridization (WISH) (*Supplementary file 1*). We also included some genes that were uniquely down-regulated in the *p53* RNAi data set (d18) because *chd4*(RNAi) animals that were to be collected for expression analysis died prematurely (before d18). A wide net was cast for candidate gene cloning, including genes with homologs predicted to be involved in a diverse array of biological processes and those with no known homologs, together spanning a wide range of expression levels. We performed an in situ expression screen to further narrow down our common gene list to those that are enriched in an *AGAT-1*-like mesenchymal or a similar epidermal pattern.

## In situ hybridization screen reveals genes enriched in *AGAT-1⁺* related cells and the gut

The majority of genes tested displayed distinctive WISH expression patterns (*Supplementary file 3*). We subsequently narrowed our candidate gene list to 29 unique genes based on a combination of their representative expression pattern, relatively strong signal intensity, and predicted gene function, for further analysis (*Figure 1E*). Based on colorimetric WISH, we classified the gene expression patterns of the 29 candidate genes into four main categories: discrete, *AGAT-1*-like sub-epidermal mesenchymal expression throughout the animal (*AGAT-1* mesenchymal); mesenchymal cells that

appeared more dense than *AGAT-1* and/or discrete expression in the epidermis (*AGAT-1* mesenchymal and epidermal); gut-enriched; and expression spanning multiple tissues exhibiting discrete mesenchymal/epidermal cells and in the gut/pharynx (*Figure 2A*). A wide assortment of biological processes were represented by the 29 candidate genes, including creatine metabolism (*AGAT-1*,

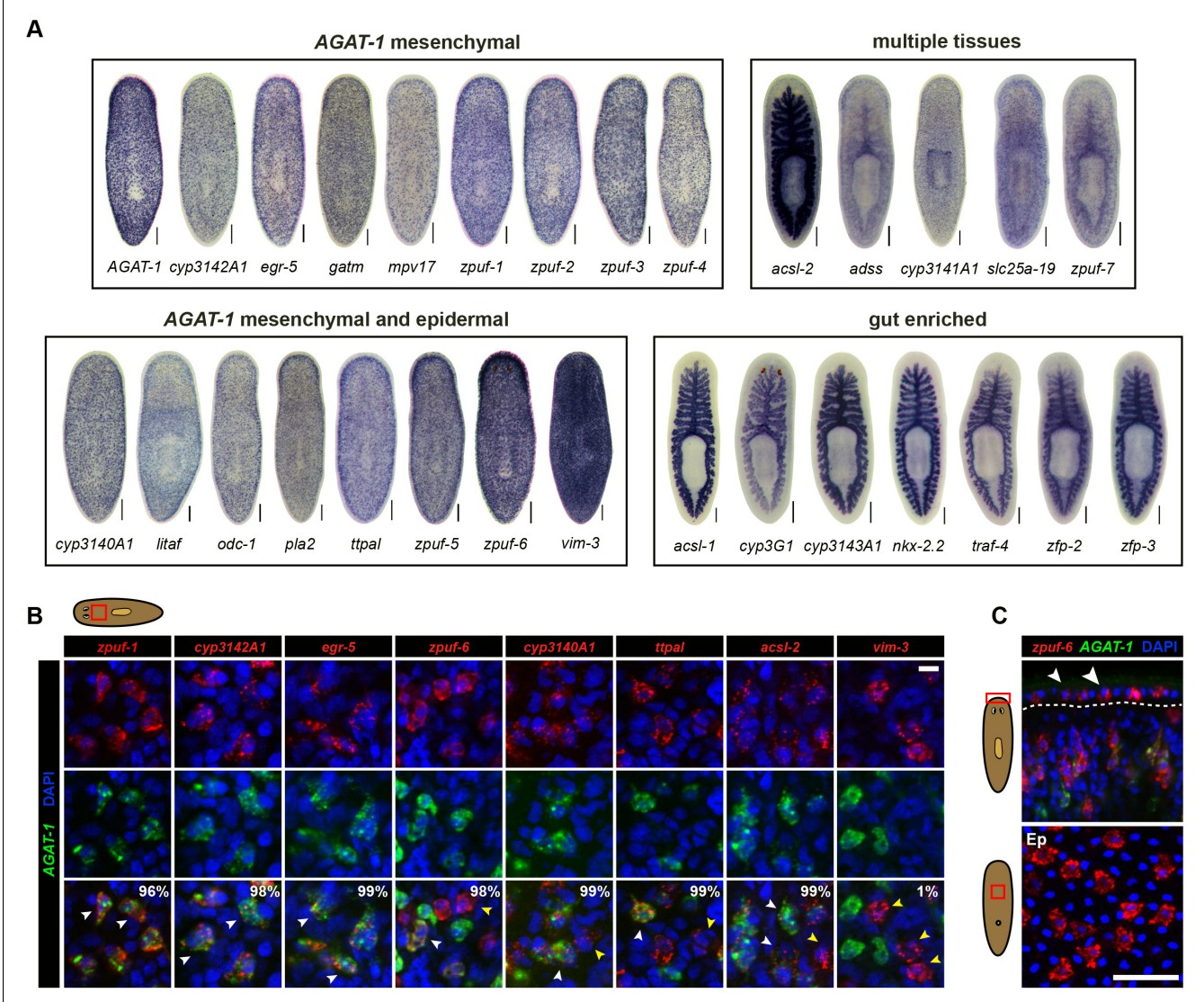

**Figure 2.** Expression patterns of candidate genes from *chd4*(RNAi) and *p53*(RNAi) data sets. (**A**) WISH expression of representative candidate gene set from *Figure 1E* and grouped by expression pattern (see text for details). Scale bars: 200 μm. See also *Supplementary file 3*. (**B**) Expression pattern of various candidate genes in (**A**), analyzed by double FISH with *AGAT-1.* Images represent single confocal planes from anterior regions. Percentages represent fraction of *AGAT-1*+ cells that co-express the candidate gene (~200-400 cells were quantified; low but detectable expression was counted as co-localized). White arrowheads highlight co-localization; yellow arrowheads highlight additional cells that have no detectable *AGAT-1* expression (*AGAT-1*^neg^). Scale bar: 10 μm. (**C**) Double FISH of *zpuf-6* and *AGAT-1* highlighting *zpuf-6* expression in discrete cells in the planarian epidermis (Top panel, white arrowheads). Bottom panel, ventral epidermal (Ep) view. DapI used for nuclear staining. Scale bar: 50 μm.

The following figure supplements are available for Figure 2:

**Figure supplement 1.** Validation of select candidate genes in *chd4, p53* and *zfp-1* RNAi conditions by WISH.

**Figure supplement 2.** Detailed characterization of *zpuf-6* expression pattern by whole-mount FISH.

**Figure supplement 3.** Expression of *pla2* and *odc-1* in mesenchyme and epidermis.

*gatm*), monooxygenase activity (*cyp3140A1, cyp3142A1, cyp3143A1, cyp3G1, cyp3141A1*), fatty-acid metabolism (*pla2, acsl-1, acsl-2*), other metabolic processes (*odc-1, mpv17, adss*), protein-binding (*traf-4*), transporter activity (*ttpal, slc25a-19*), cytoskeletal (*vim-3*), DNA-binding/zinc-finger (*egr-5, litaf, nkx-2.2, zfp-2, zfp-3*), and multiple novel genes with unknown function. These novel genes with no known homology, predicted to encode small proteins (~100 amino acids), contain a conserved signal peptide sequence. Because their expression patterns are down-regulated in *zfp-1* RNAi animals (see below), we have named them *zeta-class protein of unknown function (zpuf)*, followed by a unique designation number.

We were surprised that many candidate genes from our screen were expressed prominently in the gut, as neither *chd4* nor *p53* have been previously reported to affect intestinal gene expression. For example, *nkx-2.2* is a homeodomain TF that is enriched in the gut and has previously been shown to be required for proper intestinal regeneration (*Forsthoefel et al., 2012*). To verify the RNA-seq data by WISH, we selected a representative gene from each of the major expression categories and monitored their expression in *chd4, p53* and *zfp-1* RNAi backgrounds (*Figure 2—figure supplement 1*). The gene expression patterns for all five genes tested (*zpuf-3, zpuf-6, vim-3, nkx-2.2, acsl-2*) were significantly reduced in *chd4* and *p53* RNAi animals, whereas discrete mesenchyme and epidermal expression, but not gut expression, were reduced in *zfp-1* RNAi animals (*Figure 2—figure supplement 1A*). We conclude that transcript levels in the RNA-seq data set are highly predictive of gene expression *in vivo*.

## Candidate genes are expressed in homogeneous, overlapping *AGAT-1*[+] mesenchymal cell populations

We next performed whole-mount fluorescent in situ hybridization (FISH) to determine whether the candidate genes exhibiting discrete mesenchymal cell expression patterns overlapped with *AGAT-1* expression (*Figure 2B*). The majority of genes displayed substantial overlap with *AGAT-1*, but with varying degrees of signal intensity (*Supplementary file 3*). Several genes (*zpuf-6, cyp3140A1, ttpal* and *ascl-2*), despite showing substantial overlap with *AGAT-1* ( >98%), were also expressed in additional mesenchymal cells (*Figure 2B*, yellow arrowheads). *vim-3* was expressed in fewer sub-epidermal cells and showed very little overlap with *AGAT-1*.

After confirming that candidate genes with discrete mesenchymal cell patterns exhibit overlapping expression with *AGAT-1* by whole-mount FISH, we next focused our attention on genes that are expressed in additional *AGAT-1*-negative (*AGAT-1*[neg]) mesenchymal cells to determine any potential molecular relationships. We first characterized *zpuf-6* (also known as NB.36.10A) because it has a robust expression pattern, is expressed in *AGAT-1*[+] mesenchymal cells, and is also expressed in discrete epidermal cells (*Figure 2C*). We determined that the *zpuf-6*[+] *AGAT-1*[neg] mesenchymal cells do not overlap with *collagen* or *prog-1* (*Figure 2—figure supplement 2*), suggesting that they are neither muscle cells nor early progeny cells. Rather, these *zpuf-6*[+] *AGAT-1*[neg] mesenchymal cells likely mark a unique population linked to epidermal cells.

We conducted triple whole-mount FISH for all other candidate genes with *zpuf-6* and *AGAT-1* to resolve whether they also overlap with *zpuf-6*[+] *AGAT-1*[neg] mesenchymal cells or represent some other heterogeneous cell type. Interestingly, all genes tested displayed significant or partial overlap with *zpuf-6* in the mesenchyme and epidermis (*Supplementary file 3*). For example, *pla2* and *odc-1* both showed significant overlap with *zpuf-6*[+] *AGAT-1*[neg] mesenchymal cells and are very weakly expressed in *zpuf-6*[+] epidermal cells (*Figure 2—figure supplement 3*). Furthermore, *mpv17* expression overlaps with *zpuf-6*[+] *AGAT-1*[neg] mesenchymal cells but is not expressed in the epidermis (*Supplementary file 3*, data not shown). *Acsl-2, adss* and *vim-3* were the only genes found to be expressed in additional epidermal cell types that did not overlap with *zpuf-6*. *Adss*, which encodes the enzyme adenylosuccinate synthase and is involved in purine biosynthesis, was the one exception that appeared to be expressed in additional *AGAT-1*[neg] *zpuf-6*[neg] mesenchymal cells. Taken together, our in situ expression analysis revealed that the candidate genes identified by RNA-seq are all expressed in similar, overlapping mesenchymal and epidermal cells marked by *AGAT-1* and *zpuf-6*, and that these cell populations may all be constituents of the same lineage.

## Spatiotemporal kinetics establish lineage relationships of epidermal progeny markers

The spatiotemporal disappearance of various morphological markers *in vivo* can be used to identify cell populations that belong to the same lineage. Irradiation is a powerful tool that has been demonstrated to rapidly and specifically eliminate neoblasts, and subsequently, their immediate progeny cells (*Eisenhoffer et al., 2008*). The *prog-1*$^+$ and *AGAT-1*$^+$ cell populations were identified based on their spatial expression domains and down-regulation kinetics after irradiation. That is, the temporal order of *prog-1* (early) and *AGAT-1* (late) down-regulation correlates with the spatial distribution of these markers: the more peripheral the location of the cell, the longer the marker persists after irradiation. We were intrigued by a potential relationship between *zpuf-6* and *vim-3*, also identified in our screen (*Figure 2*), because it is expressed in fewer sub-epidermal mesenchymal cells that do not overlap with *AGAT-1*, but more epidermal cells than *zpuf-6*. Therefore, both *zpuf-6* and *vim-3* from our candidate gene set expand the spatial distribution of known progeny markers into the epidermis (*Figure 3A*). The distinct expression domains of *prog-1, AGAT-1, zpuf-6,* and *vim-3* suggest that these progeny cells undergo outward migration during the course of epidermal differentiation.

We therefore extended this paradigm, that more differentiated cells have slower turnover kinetics, to determine the spatiotemporal irradiation kinetics of these newly characterized markers. After irradiation, both *zpuf-6* and *vim-3* displayed slower down-regulation kinetics compared to *AGAT-1* and exhibited expression loss in a similar ventral-to-dorsal and anterior-to-posterior manner (*Figure 3B*, *Figure 3—figure supplement 1*). We also monitored the expression of *rootletin*, a gene expressed in the ciliated epidermis and tubule cells of the protonephridia (*Glazer et al., 2010*; *Scimone et al., 2011*), as a marker for differentiated cells and confirmed that its expression is not significantly affected after irradiation (*Figure 3B*, bottom panel). Therefore, the irradiation kinetics data support our interpretation that *zpuf-6* and *vim-3* mark post-mitotic cells further downstream of the same *prog-1* and *AGAT-1* epidermal lineage. In other words, *AGAT-1*$^+$ cells likely give rise to *zpuf-6*$^+$ and *vim-3*$^+$ expressing cells.

We also looked for a conservation in *AGAT-1* and *zpuf-6* spatiotemporal expression kinetics in additional contexts. In both *chd4* and *p53* RNAi backgrounds, *AGAT-1*$^+$ and *zpuf-6*$^+$ cells are progressively lost in a ventral-to-dorsal, anterior-to-posterior fashion, with *zpuf-6*$^+$ cells also undergoing a more delayed loss compared to *AGAT-1* (*Figure 3—figure supplement 2A*). Interestingly, in *p53* (RNAi) animals, there is a characteristic banding pattern of *AGAT-1*$^+$ and *zpuf-6*$^+$ cells on the ventral surface of the animals (*Figure 3—figure supplement 2A* bottom panels, 2B) as cells are progressively lost over the course of RNAi treatment. This banding pattern of cells is also present for *vim-3* on the ventral epidermis in *p53*(RNAi) animals at later time points (not shown), further suggesting that *AGAT-1, zpuf-6,* and *vim-3* are markers exhibiting strong spatial correlative patterns.

## *zpuf-6*$^+$ epidermal cells mark a transitional rather than differentiated state

Lineage tracing experiments using the thymidine analog bromodeoxyuridine (BrdU), in combination with whole-mount FISH, have revealed spatial and temporal regulation of neoblasts and the distribution of their differentiating progeny cells (*Eisenhoffer et al., 2008*; *Newmark and Sánchez Alvarado, 2000*). A single-pulse of BrdU has also revealed the turnover dynamics of *prog-1*$^+$ and *AGAT-1*$^+$ and epidermal cells, revealing that *prog-1* and *AGAT-1* become incorporated markedly earlier than epidermal cells, consistent with the notion that they mark an intermediate stage of epidermal differentiation (*van Wolfswinkel et al., 2014*).

We performed BrdU pulse-chase analysis to examine the temporal kinetics of BrdU-labeling from neoblasts to *prog-1*$^+$, *AGAT-1*$^+$ and *zpuf-6*$^+$ cells during normal tissue turnover. The percentage of *prog-1*$^+$ BrdU$^+$ cells reached maximum levels around 10 days post-BrdU, followed by *AGAT-1*$^+$ BrdU$^+$ cells peaking at 14 days post-BrdU, and *zpuf-6*$^+$ BrdU$^+$ cells reached maximum levels at 22 days post-BrdU (*Figure 4A*). Confidence tests of the difference between individual data points show that through days 1—14, the difference between *AGAT-1*$^+$ BrdU$^+$ and *zpuf-6*$^+$ BrdU$^+$ are only marginally significantly different (most significant data point p = 0.04 for d6). However, at 22 days post-BrdU the difference between the points of these same curves is highly significant (p <0.01). These

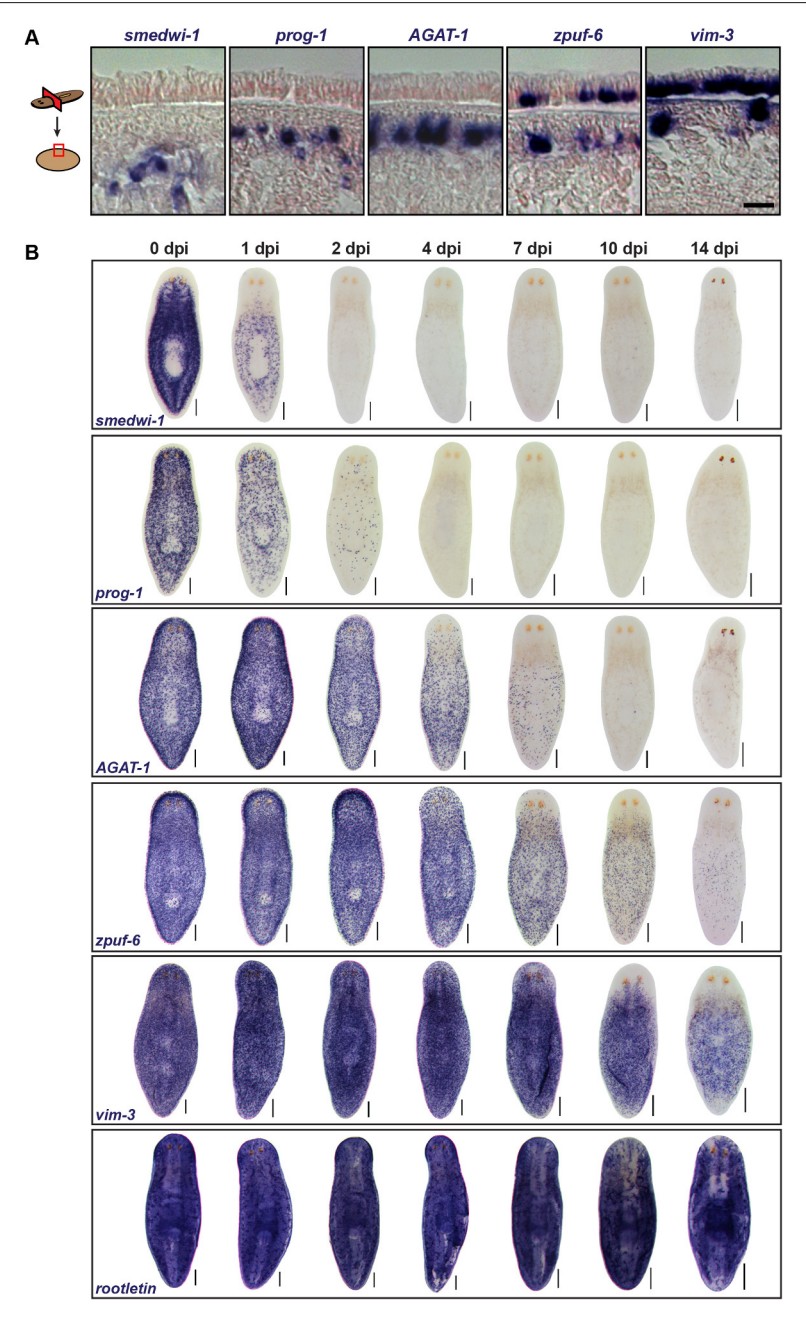

**Figure 3.** Epidermal progeny markers are expressed in distinct spatiotemporal domains. (**A**) Transverse tissue sections of colorimetric WISH-stained animals expressing putative markers in the epidermal lineage. Sections were counterstained with nuclear fast red. Scale bar: 10 μm. (**B**) Colorimetric WISH of animals after 6000 Rad irradiation exposure for markers of neoblasts (*smedwi-1*), early progeny (*prog-1*), late progeny (*AGAT-1*), and other genes identified in this study (*zpuf-6* and *vim-3*) marking transitions into the epidermis. *Rootletin* marks differentiated cells of the ciliated epidermis as well as protonephridia. Expression patterns of epidermal progeny markers are lost in a ventral-to-dorsal, anterior-to-posterior manner. A representative image from 4–6 worms for each gene and time point are shown. dpi, days post-irradiation. Dorsal views. See *Figure 3—figure supplement 1* for close-up ventral views. Scale bars: 200 μm.

The following figure supplements are available for Figure 3:

**Figure supplement 1.** Ventral plane sections of epidermal progeny markers after irradiation.

*Figure 3. continued on next page*

*Figure 3. Continued*

**Figure supplement 2.** Correlated spatial expression patterns of *prog-1, AGAT-1* and *zpuf-6*.

results lend further support to our hypothesis that *prog-1, AGAT-1,* and *zpuf-6* are distinct markers representing progressive stages of epidermal progeny differentiation.

Because *zpuf-6* labels the first transition state into the planarian epidermis, we wondered whether *zpuf-6*[+] epidermal cells embody a specific differentiated epidermal cell type or whether they still possess the potential to differentiate further. To address this question, we built upon the spatial and molecular relationships between *AGAT-1, zpuf-6* and *vim-3* defined by our previous irradiation kinetic studies by performing combinatorial whole-mount FISH. Triple whole-mount FISH of *AGAT-1, zpuf-6* and *vim-3* revealed that *vim-3* overlaps with *AGAT-1*[neg]*zpuf-6*[+] mesenchymal cells, *zpuf-6*[+] epidermal cells, and there are additional *zpuf-6*[neg]*vim-3*[+] epidermal cells present on the dorsal side of animals (**Figure 4B**). *vim-3*, predicted to encode an intermediate-filament like protein, also co-localizes with *vim-1* (**Figure 4—figure supplement 1A**), which was shown to be down-regulated in *zfp-1* RNAi animals (**van Wolfswinkel et al., 2014**). Intermediate filaments comprise a diverse class of molecules and are expressed in a variety of cell types, including epithelial cells. These filaments generally provide a scaffold to integrate components of the cytoskeleton and organize the internal cell structure (**Snider and Omary, 2014**). Thus, we wondered whether the expression overlap of *zpuf-6* and *vim-3* could reflect cells undergoing a morphological transition within the epidermis.

The formation of cilia is a signature of terminal differentiation (**May-Simera and Kelley, 2012**). The ventral epidermis is lined with multi-ciliated cells responsible for gliding motility, while the more densely packed dorsal surface contains many non-ciliated and mucous-secreting, rhabdite-containing cells (**Pedersen, 1976**). Given that *zpuf-6* expression is evenly distributed throughout the dorsal and ventral epidermis, we queried whether *zpuf-6*[+] epidermal cells co-expressed markers for cilia genes. We used the planarian *rootletin* gene as a marker for ciliated epidermal cells and found that *zpuf-6* and *rootletin* do not significantly overlap in expression (**Figure 4C**, top row), suggesting that *zpuf-6*[+] epidermal cells are non-ciliated. However, the combination of *vim-3* and *rootletin* in whole-mount FISH reveals that while cells expressing high levels of *vim-3* display virtually undetectable levels of *rootletin*, there are cells with detectable *vim-3* expression that do co-express *rootletin* (**Figure 4C**, bottom row), suggesting that *vim-3* cells may begin to undergo ciliogenesis.

## *zpuf-6* expression in the regenerating blastema and other epidermal cell types

Under normal homeostasis conditions, *zpuf-6*[+] epidermal cells appear randomly but broadly distributed on both the dorsal and ventral epidermal surface of intact animals (**Figure 2**). We reasoned that if *zpuf-6*[+] epidermal cells still harbor the potential to differentiate further, then the density and distribution of *zpuf-6*[+] cells would increase if epidermal integrity were severely perturbed. Therefore, we amputated wild-type animals and examined *zpuf-6* expression in the regenerating blastema over 7 days (**Figure 4—figure supplement 2**). Consistent with our hypothesis, the density of *zpuf-6*[+] epidermal cells as measured by fractional area is higher in the new undifferentiated blastema tissue compared to old tissue at 4 and 7 days post-amputation.

We also used whole-mount FISH to examine *zpuf-6* expression with *NB.22.1E* and *laminB*, markers that are expressed in specific domains in the epidermis (**van Wolfswinkel et al., 2014**). *NB.22.1E* labels marginal adhesive gland cells at the body margin of the animal (**Reddien et al., 2007**; **Tazaki et al., 2002**) as well as cells lining the ventral mouth opening. Curiously, there was very little overlap between *zpuf-6* and *NB.22.1E* at the edge, but *NB.22.1E*[+] cells around the ventral opening co-expressed *zpuf-6* (**Figure 4—figure supplement 3A**). *LaminB* is also expressed mainly in marginal adhesive cells and a few subepidermal cells near the edge, and displayed little overlap with *zpuf-6* (**Figure 4—figure supplement 3A**). We wondered whether there was minimal co-expression between *zpuf-6* and *NB.22.1E* and *laminB* because these marginal adhesive cells do not turnover as quickly as epidermal cells on the dorsal and ventral surface. To test this, we monitored the kinetics of *NB.22.1E* and *laminB* expression after a lethal dose of irradiation. After 14 days post-irradiation, *NB.22.1E* and *laminB* mesenchyme expression is completely lost, but only cells in the anterior region have disappeared, indicating that the marginal edge cells do not turnover as quickly (**Figure 4—**

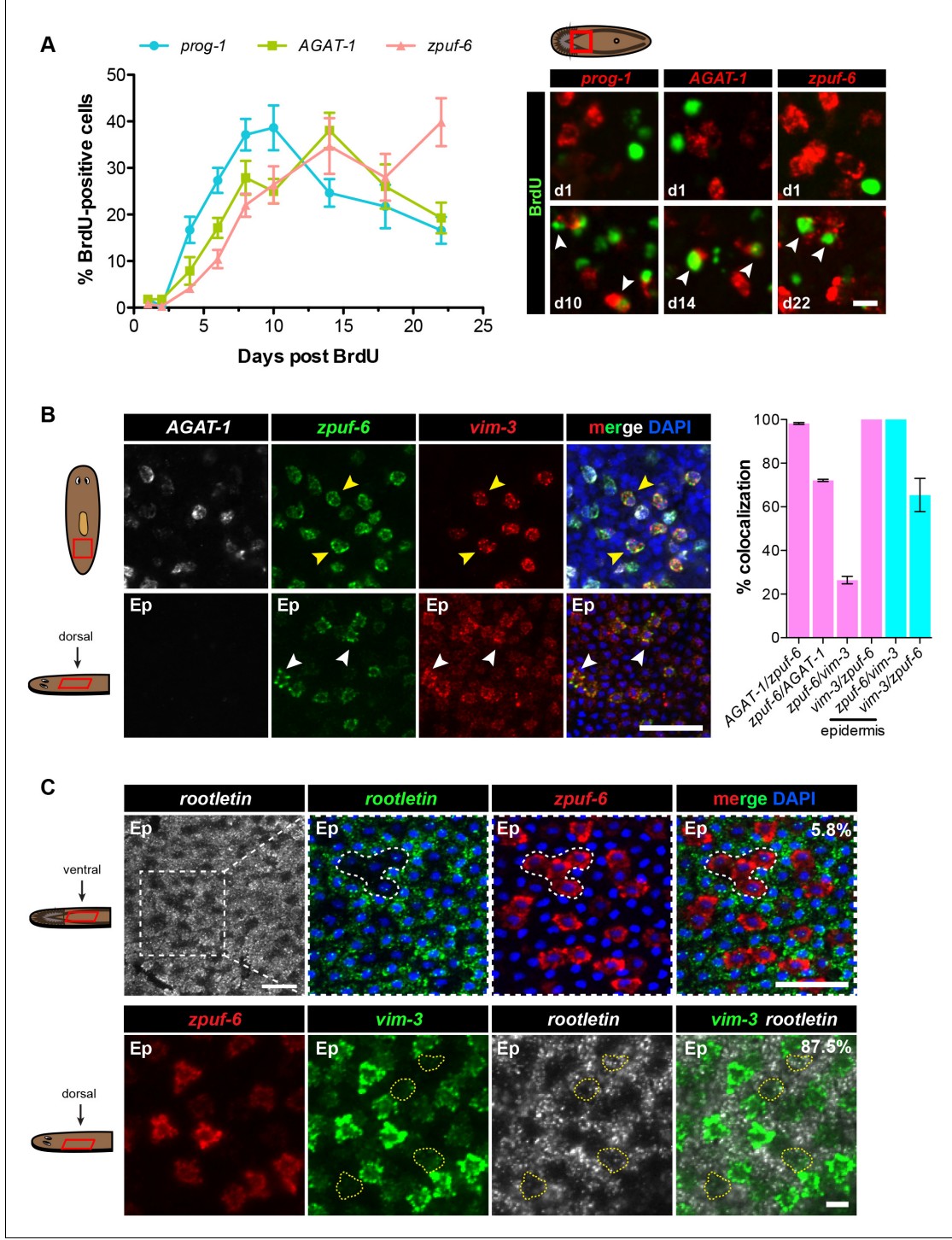

**Figure 4.** *Zpuf-6*+ epidermal cells are not terminally differentiated. (**A**) Turnover dynamics for *prog-1, AGAT-1* and *zpuf-6* expressing cell populations. Animals were soaked with BrdU for 24 hr and chased for the indicated time periods. Quantification of the percentage of *prog-1*+, *AGAT-1*+ or *zpuf-6*+ cells analyzed that are BrdU+ are plotted. Error bars: SEM (see Materials and methods). Representative images of BrdU+ cells, are maximum intensity projections over 1 cell diameter of a subset of the quantified images, of the minimum and maximum time points for each gene are shown. White arrowheads highlight double-positive cells. Scale bar: 10 μm. (**B**) Lineage relationship between *AGAT-1, zpuf-6* and *vim-3*. Top row: Triple FISH showing overlapping expression between *AGAT-1, zpuf-6* and *vim-3* in the mesenchyme. Yellow arrowheads highlight cells that are *zpuf-6*+ *vim3*+ *AGAT-1*neg. *AGAT-1* and *vim-3* exhibit little to no co-expression (**Figure 2B**). Bottom row: Dorsal epidermal (Ep) view of

*Figure 4. continued on next page*

*Figure 4. Continued*

*zpuf-6* and *vim-3*. White arrowheads highlight cells that co-express *zpuf-6* and *vim-3* but there are additional cells expressing *vim-3*. Images are single confocal planes. Scale bar: 50 μm. Quantifications of the percent colocalizations of combinations of *AGAT-1, zpuf-6* and *vim-3* mesenchymal (magenta) and *zpuf-6* and *vim-3* dorsal epidermal cells (cyan) are shown (~200-400 cells were quantified over >3 animals; error bars: SD). *Gene1/gene2* notation signifies percentage of *gene1*+ cells that are also positive for *gene2* expression. Notable percentages: *zpuf-6*+/*AGAT-1*+ (72%), *zpuf-6*+/*vim-3*+ (26%). **(C)** *zpuf-6+* epidermal cells express very low levels of *rootletin*. Top row: double FISH of *rootletin* and zpuf-6. 5.8%=percentage of strong *zpuf-6+* epidermal cells that co-express *rootletin* (~300 cells). Scale bars: 50 μm. Bottom row: triple FISH of *zpuf-6, vim-3* and *rootletin* in dorsal epidermis. Only overlay of *vim-3* and *rootletin* is shown (far right panel). Yellow-dashed shapes outline cells that express very low levels of *vim-3* that also express *rootletin*. 87.5%=percentage of weak *vim-3+* dorsal epidermal cells that co-express *rootletin* (~200 cells). Images are single confocal planes. Scale bar: 10 μm.

The following figure supplements are available for Figure 4:

**Figure supplement 1.** *Vim-3* and *vim-1* are co-expressed in the same cell types.

**Figure supplement 2.** *zpuf-6*+ cells are enriched at regenerating blastemas.

**Figure supplement 3.** *NB.22.1E*+ and *laminB*+ cells exhibit little overlap with *zpuf-6* and display slow cell turnover kinetics after irradiation.

---

*figure supplement 3B*). Moreover, *vim-3* overlaps considerably with both *NB.22.1E*+ and *laminB*+ cells at the animal body margin (*Figure 4—figure supplement 1B*), suggesting that both *zpuf-6* and *vim-3* are expressed in multiple epidermal cell types.

Taken altogether, transcriptional profiling of *chd4* and *p53* RNAi animals has identified additional epidermal progeny gene markers that are expressed broadly throughout the animal in discrete but overlapping cell populations in the mesenchyme and epidermis. *zpuf-6*+ epidermal cells can still undergo progressive differentiation into an unknown number of epidermal cell types (potentially including *NB.22.1E*+ cells lining the mouth and *laminB* marginal adhesive cells), by expressing markers of cytoskeletal morphogenesis (*vim-1/vim-3*) followed by cilia gene markers (*rootletin*). The addition of these new molecular markers in the planarian epidermal lineage will greatly facilitate the study of epidermal progenitor dynamics and differentiation during tissue homeostasis and regeneration.

## *Egr-5* is a conserved TF expressed in epidermal progenitor cells

Transcriptional changes are likely drivers of maturation within the epidermal lineage, which occurs in the absence of cell division. We therefore focused on characterizing putative transcription factors from our common down-regulated gene set as likely candidates important for post-mitotic epidermal differentiation. *egr-5* is a planarian homolog of the early growth response (*EGR1*) family of C2H2-type zinc-finger TFs (*Figure 5—figure supplement 1*). *EGR1* genes are known to be induced by extracellular signals including growth factors, hormones and neurotransmitters, and couple these signals to long-term responses by altering the expression of target genes (*Thiel and Cibelli, 2002*). *Egr-5* is expressed in discrete mesenchymal cells located throughout the animal in a spatial pattern similar to *AGAT-1* cells (*Figure 2B*). We examined *egr-5* expression levels in greater detail by whole-mount FISH in combination with other post-mitotic markers of the epidermal lineage. Robust *egr-5* expression appears to correlate with strong *AGAT-1* expression, but *egr-5* is also lowly expressed in *AGAT-1*$^{neg}$ *zpuf-6*+ mesenchymal cells (*Figure 5—figure supplement 2A*, top panel). By whole-mount FISH metrics, *egr-5* expression is also barely detectable in a subset of *prog-1*+ cells and in the epidermis (*Figure 5—figure supplement 2A*, bottom panels), demonstrating that *egr-5* expression, although varied in signal intensity, spans the domain of all post-mitotic lineage markers. In addition, *egr-5* transcripts are predominantly found in the FACS-dissociated Xins population comprised of post-mitotic cells and not in the X1 neoblast dividing fraction (*Figure 5—figure supplement 2B*), further suggesting that *egr-5* mainly functions in post-mitotic cells.

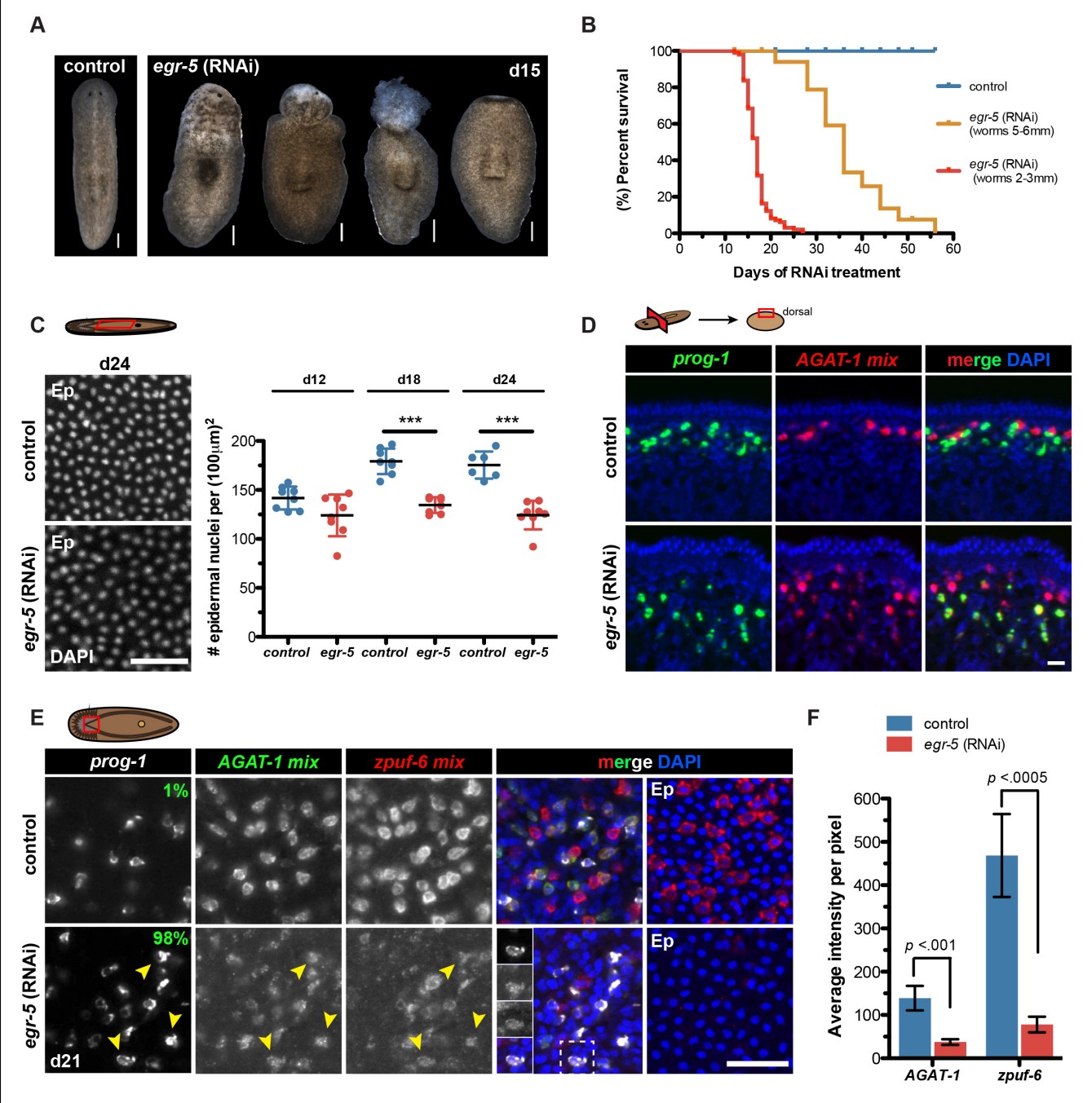

**Figure 5.** *Egr-5* is required for the proper differentiation of epidermal progeny cells. (**A**) Intact phenotypes for representative control and *egr-5*(RNAi) animals. Various stages of phenotypic progression of *egr-5* RNAi knockdown (4Fd15) are shown. Scale bars: 200 μm. (**B**) Survival curves for control and *egr-5*(RNAi) animals. The efficacy of RNAi and resulting gross phenotype is dependent on size of animals at the start of RNAi feedings. Animals starting around 2-3 mm in size (red lines) were fed 5 times (5F) (n = 99); animals starting around 5-6 mm in size (yellow lines) were fed 6 times (6F) (n = 65). Death was marked by complete animal lysis. (**C**) Effects of *egr-5*(RNAi) on epidermal cell density. DapI was used to quantify epidermal (Ep) nuclear density in the ventral mid-sections; representative images for 4Fd24 are shown. Scale bar: 50 μm. Quantification of epidermal cell density are plotted for d12, d18 and d24 for control (blue) and *egr-5*(RNAi) (red) animals. Each symbol represents individual animals (average of 2 regions/animal). Black lines and error bars (colored) represent mean and SD, respectively. Student's t test: ***, p <0.00005. (**D**) Spatial domain expansion of *prog-1* and *AGAT-1* cells in *egr-5*(RNAi). Tissue transverse sections of control and *egr-5*(RNAi) animals stained for markers of early epidermal progeny cells (8Fd30). To help improve visualization of *AGAT-1*-expressing cells, an RNA probe mix of other *AGAT-1* co-expressed genes were used (*AGAT-1* mix = *AGAT-1*, *zpuf-1*, *zpuf-3*, *zpuf-4*). Scale bar: 10 μm. (**E**) Misexpression of epidermal progeny markers in *egr-5*(RNAi) animals. Triple FISH of *prog-1*, *AGAT-1* and *zpuf-6* markers in control and *egr-5*(RNAi) animals (4Fd21). Percentages shown in *prog-1* panel represent percentage of *prog-1*[+] cells that also express

*Figure 5. continued on next page*

*Figure 5. Continued*

AGAT-1 (400—700 cells were counted between 3—4 worms per condition). Yellow arrowheads highlight cells that are *prog-1*[+] *AGAT-1*[+] *zpuf-6*[+] (also shown in inset panels in merge panel). Rightmost column: epidermal (Ep) view highlighting loss of *zpuf-6* expression in epidermis. AGAT-1 mix = AGAT-1, zpuf-1, zpuf-3, zpuf-4; zpuf-6 mix = zpuf-6, zpuf-5 and zpuf-8. Scale bar: 50 μm. (**F**) Average fluorescence intensity for *AGAT-1* mix and *zpuf-6* mix probes in control and *egr-5*(RNAi) animals from (**E**). Error bars: SD. p-values are results of unpaired Student's t test.

The following figure supplements are available for Figure 5:

**Figure supplement 1.** Smed-Egr-5 is a conserved member of the early growth response family of transcription factors.

**Figure supplement 2.** *Egr-5* is expressed in multiple post-mitotic epidermal progeny cells.

**Figure supplement 3.** Molecular and ultrastructural analysis of *egr-5*(RNAi) epidermis.

**Figure supplement 4.** Reduction of *laminB*[+] cells in *egr-5*(RNAi) animals.

## *Egr-5* is a regulator of post-mitotic epidermal lineage fate specification

To investigate the role of *egr-5* during normal tissue turnover, animals were fed *egr-5* dsRNA every 3 days and subsequently screened for gross morphological defects. *egr-5*(RNAi) animals displayed a range of deformities: loss of epidermal integrity, anterior blebbing, and complete loss of the anterior region (**Figure 5A**). Moreover, the severity of phenotypic progression leading to animal death caused by lysis is inversely correlated with animal size (**Figure 5B**). Because these gross morphological defects manifested in *egr-5*(RNAi) animals suggested problems with epidermal morphology or density, we quantified epidermal nuclear density and found that *egr-5*(RNAi) animals had a significantly reduced number of epidermal cells compared to control animals (**Figure 5C**).

Next, we investigated epidermal morphology by looking at expression of *zpuf-6* and *rootletin* using whole-mount FISH and ultrastructural features by scanning electron microscopy (SEM). We observed a significant reduction in *zpuf-6* expression (see more below) in the epidermis between day 12 and day 24 of RNAi treatment, whereas *rootletin* expression did not diminish, although it appeared disorganized compared to controls (**Figure 5—figure supplement 3A–B**). Analysis of the ventral cilia by SEM qualitatively showed similar levels in abundance. However, the ventral epidermal cells in *egr-5*(RNAi) animals appeared more smooth and stretched out and lacked epidermal pores/pits as seen in control animals (**Figure 5—figure supplement 3C**). The smoothness of the ventral epidermis also contributed to difficulty in delineating individual epidermal cells. Our data suggest that the reduced epidermal density in *egr-5*(RNAi) animals could be caused by the failure of new epidermal progeny cells to properly differentiate and incorporate into the epidermis, resulting in stretching and apparent morphological defects. Consistent with this hypothesis, we also found that there was a significant reduction in the number of *laminB*[+] cells in the anterior region of *egr-5*(RNAi) animals compared to control animals after 21 days of RNAi treatment (**Figure 5—figure supplement 4**).

## Misexpression of epidermal progeny markers in *egr-5*(RNAi) animals

Given that *egr-5* knockdown causes a marked reduction in *zpuf-6* expression in the epidermis, we looked at our panel of epidermal lineage markers to determine which particular transition may be affected in the course of differentiation. In RNAi conditions such as *chd4, p53* and *zfp-1* knockdown where *AGAT-1*[+] cell populations are lost, there is a distinct progressive disappearance in the spatial distribution of cells, but the signal intensity of the gene marker expressed in the remaining cells is not significantly diminished (see **Figure 2—figure supplement 1A**; **Figure 3—figure supplement 2B**). This suggests that these factors are required for the generation or maintenance of *AGAT-1*[+] cells, but not necessarily required for the expression of *AGAT-1* or other markers in these cells. However, a TF can be responsible for the expression of *AGAT-1* or some other marker, and not be required for the maintenance of that cell type, or both. To distinguish between the loss of cell type versus the loss of specific gene expression, we made RNA probe mixes consisting of multiple *zpuf* genes from our screen expressed in either *AGAT-1* or *zpuf-6* cells to increase overall signal intensity and detection.

We first investigated the spatial distribution of *prog-1* and *AGAT-1* progeny markers and found that the stereotypical sub-epidermal expression domains of both markers had expanded deeper into the mesenchyme in *egr-5*(RNAi) animals (*Figure 5D*), suggesting potential temporal defects in the earlier steps of epidermal lineage progression. We then simultaneously analyzed *prog-1, AGAT-1* and *zpuf-6* markers by whole-mount FISH and established that in *egr-5*(RNAi) animals, epidermal progeny cells in the mesenchyme are not specifically lost, but that *AGAT-1* and *zpuf-6* expression levels were significantly reduced (*Figure 5E–F*). Although *zpuf-6* mesenchymal expression is very weak after *egr-5* knockdown, it is virtually undetectable in the epidermis, potentially highlighting a key transitional defect. Given that *egr-5* knockdown reduces both epidermal cell density as well as the expression of *AGAT-1* and *zpuf-6* markers, we suggest that *egr-5* is responsible for both the maturation of epidermal progeny cells and the expression of progeny markers. Interestingly, *prog-1* and *AGAT-1* normally do not exhibit significant overlapping expression domains, but here they display significant overlapping expression patterns after loss of *egr-5* (*Figure 5E*, left panels). Together, our results demonstrate an essential role for *egr-5* in the proper spatial and temporal progression of progeny cells in the epidermal lineage and that misexpression of these marker domains likely causes improper differentiation and failed epidermal maturation.

## Expansion of neoblasts and multiple progenitors in *egr-5*(RNAi) animals

Although *egr-5* is expressed in post-mitotic epidermal lineage cells to coordinate their proper temporal transition states, we wondered if loss of *egr-5* may have any non-autonomous effects on neoblast and progenitor dynamics. To examine neoblast division kinetics, we quantified histone H3Ser10 phosphorylation (H3P) during *egr-5* RNAi knockdown and observed a significant increase in neoblast proliferation, followed by an eventual decline (*Figure 6A–B*).

Because neoblasts constitute a mixed population of pluripotent stem cells and lineage-committed progenitors, we looked more closely at the composition of dividing neoblasts to determine if *egr-5* knockdown expanded all dividing progenitors or only affected a specific subset. We picked a time point where H3P levels were high in *egr-5*(RNAi) animals (before the decline) and then analyzed the proportion of dividing progenitors for major tissues, including the epidermis (zeta-class) (*van Wolfswinkel et al., 2014*), protonephridia (*pou2/3*) (*Scimone et al., 2011*), gut (*hnf4*) (*Wagner et al., 2011*) and brain (*pax6A*) (*Wenemoser et al., 2012*) (*Figure 6C–G, Figure 6—figure supplement 1A*). Despite demonstrating a dramatic increase in overall dividing progenitors, *egr-5* (RNAi) animals displayed similar proportions of dividing tissue progenitors compared to their control counterparts. These results suggest that *egr-5* RNAi knockdown causes an overall expansion of multiple lineage progenitors, including the epidermal progenitors (zeta-class). Therefore, failure to generate mature epidermal cells in *egr-5*(RNAi) animals is not caused by a failure to generate zeta-class epidermal progenitors. Rather, it is likely caused by a defect at the post-mitotic differentiation level. This decrease over time in the integration of functional, differentiated cells leads to the eventual breakdown of epidermal integrity and results in animal lysis.

## Differentiation defects in *egr-5*(RNAi) animals are specific to epidermal lineages

To assess whether the expanded progenitor population in *egr-5*(RNAi) animals led to any pronounced defects in differentiation, we examined by colorimetric WISH the protonephrida (*slc6a-13*) (*Vu et al., 2015*), epidermal progenitors/*prog-1/AGAT-1* (*p53*), and overall gut morphology (*hnf4*). Qualitatively, all markers exhibited prominent increases in number and expression in *egr-5*(RNAi) animals compared to controls (*Figure 6—figure supplement 1B–D*). We quantified the number of protonephridial proximal units (PU) and found that *egr-5*(RNAi) animals indeed display a significant increase in protonephridial density (*Figure 6H–I*). Based on general morphology and the observation that *egr-5* RNAi animals did not exhibit defects in osmoregulation by bloating, we conclude that the protonephridia are functional. The strong increase in *p53* expression builds upon our previous finding that loss of *egr-5* increases the number of zeta-class epidermal progenitors (*Figure 6C, Figure 6—figure supplement 1A*), as well as *prog-1* and *AGAT-1* cells (*Figure 5D–E*). The enriched gut expression marked by *hnf4* in *egr-5*(RNAi) animals also suggests the presence of an increased number of gut cells, without any discernible physiological consequences (*e.g.*, animals can still eat). Together, our data suggest that the extensive increase in neoblast proliferation caused by *egr-5*

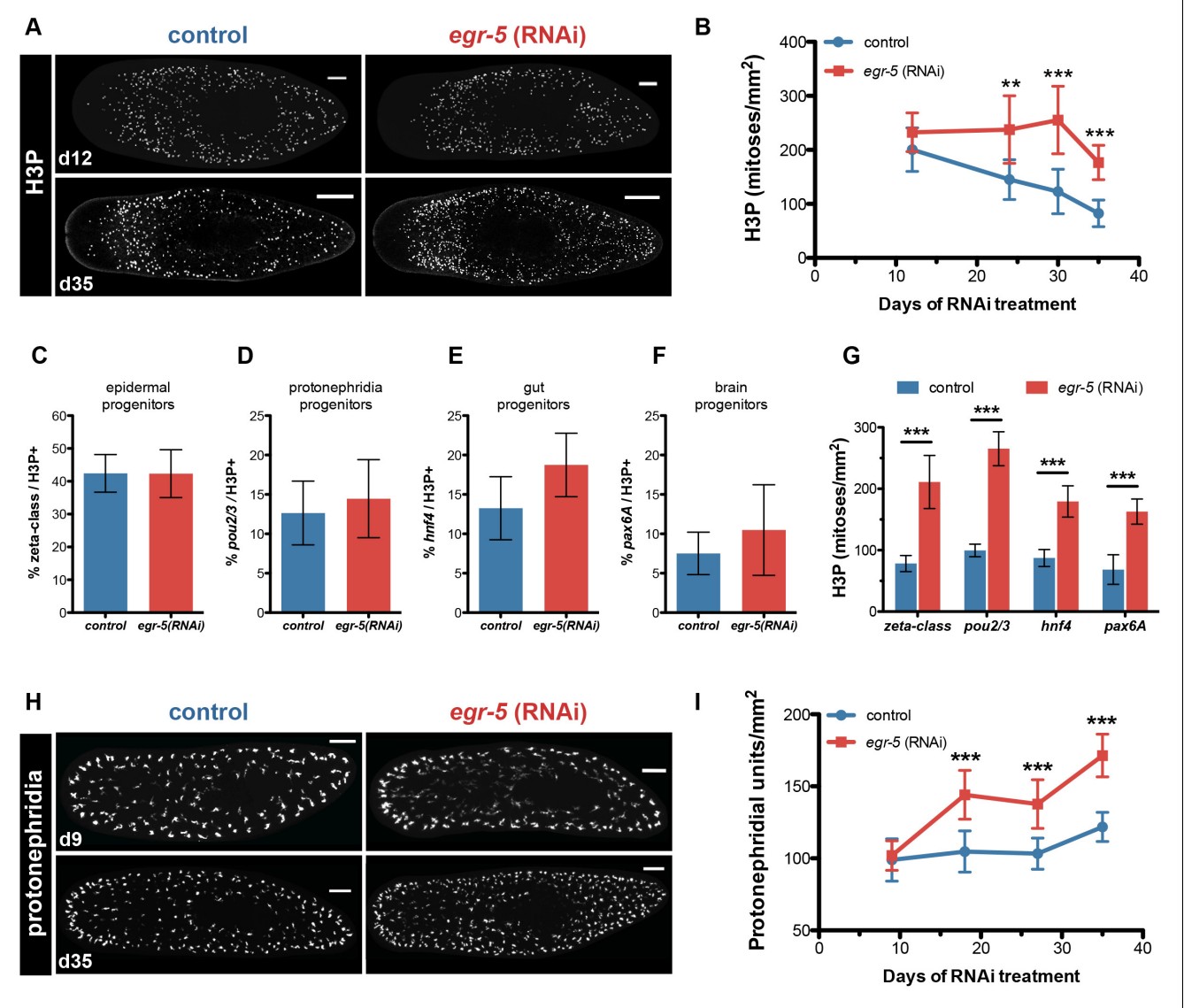

**Figure 6.** *Egr-5* knockdown results in the expansion of multiple progenitor populations. (**A,B**) *egr-5* knockdown causes an increase in global stem cell proliferation. (**A**) H3P-positive cells over the surface area in control and *egr-5*(RNAi) intact animals (8 feedings). Maximum intensity projections of representative H3P patterns for day 12 and day 35 time points are shown. Scale bar: 200 μm. (**B**) Quantification of H3P-positive cells per surface area from (**A**). Data represent means, error bars: SD. Student's t test: **, $p < 0.005$, ***, $p < 0.0001$. (**C–G**) Stem cell proliferation results in the expansion of multiple lineage-committed progenitors. For control and *egr-5*(RNAi) animals (3Fd27), animals were fixed and stained for H3P and a marker for the following: (**C**) dividing epithelial progenitors (zeta class: *zfp-1, egr-1, fgfr-1*), (**D**) dividing protonephridia progenitors (*pou2/3*), (**E**) dividing gut progenitors (*hnf4*), and (**F**) dividing brain progenitors (*pax6A*). Quantifications of percentage of lineage-committed progenitors of total H3P$^+$ cells are shown. Error bars: SD. (**G**) Quantification of total mitotic cells (H3P) per surface area for control and *egr-5*(RNAi) animals (3Fd27) from analysis in (**C–F**). Data represent means, error bars: SD. Student's t test: ***, $p < 0.0001$. (**H,I**) *egr-5* knockdown causes supernumerary protonephridial units (PU). (**H**) Control and *egr-5*(RNAi) intact animals (8 feedings) were stained for the number of proximal units (*slc6a-13*) to visualize and quantify total PUs. Maximum intensity projections of representative animals for day 9 and day 35 time points are shown. Scale bar: 200 μm. (**I**) Quantification of PU per surface area. Data represent means, error bars: SD. Student's t test: ***, $p < 0.0001$.

The following figure supplements are available for Figure 6:

**Figure supplement 1.** Analysis of multiple dividing tissue progenitors in *egr-5*(RNAi) animals.

depletion is remarkably counterbalanced by a corresponding increase in the normal progression of multiple lineage progenitors that differentiate and integrate into functional tissues.

## Loss of *egr-5* induces a global stress response

The potential non-cell autonomous effect of *egr-5* knockdown on neoblast and progenitor dynamics raised the possibility that epidermal defects resulting from abnormal differentiation and defective homeostasis may induce some kind of global stress response. Therefore, we measured apoptosis using a whole-mount TUNEL assay (*Pellettieri et al., 2010*) and observed a remarkable increase in cell death throughout the course of *egr-5* RNAi treatment compared to control animals (*Figure 7A*). Notably, the increase in cell death appeared to occur uniformly throughout the animal, with the majority of TUNEL-positive nuclei found in the mesenchymal tissue and not in the epidermis (not shown).

A global increase in both cell proliferation and cell death are general features of the initial stages of planarian regeneration (*Pellettieri et al., 2010*; *Wenemoser and Reddien, 2010*). Simple wounding and amputations causing loss of tissue are both capable of activating the expression of many wound-induced genes within the first 24 hr of insult (*Wenemoser et al., 2012*). We hypothesized that the epidermal defects caused by *egr-5* RNAi knockdown may trigger the activation of wound-induced genes expressed in differentiated cells. *Delta-1* is a putative Notch signaling pathway ligand whose expression is induced in the epidermis between 6-24 hr after injury, not necessarily localized to the site of wounding (*Wenemoser et al., 2012*). We monitored the expression of *delta-1* in intact, uninjured *egr-5*(RNAi) animals and observed a marked increase in *delta-1* in the anterior region of animals compared to their control counterparts through 21 days of RNAi treatment (*Figure 7B*, top row). We also measured the in vivo expression of other 'immediate early genes' that are activated after wounding including *fos-1, jun-1* and *egr-3* (*Wenemoser et al., 2012*), and observed notable increases in the anterior region of *egr-5*(RNAi) animals (*Figure 7B*, middle rows). Interestingly, the TGF-β inhibitor *follistatin*, whose expression is specifically induced for multiple days after a loss of animal tissue (*Gavino et al., 2013*), does not appear to be stimulated over the course of *egr-5* RNAi treatment (*Figure 7B*, bottom row). In addition, the immediate early genes *fos-1, jun-1* and *egr-3* are enriched in our *chd4* and *p53* RNAi whole-worm RNA-seq up-regulated datasets (*Supplementary file 2*; *Figure 7—figure supplement 1*). Taken together, these data suggest that both defects in epidermal differentiation (*egr-5* RNAi) and loss of epidermal progeny cells (*chd4* and *p53* RNAi) likely contribute to a breach in epidermal integrity, which consequently may activate a systemic wound response program.

## Discussion

As the only dividing somatic cells in planarians, neoblasts must sense the environment and respond accordingly to meet the demands of the organism. It is not well understood how neoblasts elaborate specific programs of post-mitotic differentiation to generate mature tissue cell types. Here, we provide further evidence that the planarian epidermis is an experimentally tractable system to study the complex dynamics and hierarchical transitions that are likely to occur during adult lineage specification and progression.

## Planarian epidermal maturation requires multiple transition states

Zeta-class neoblasts, marked by *zfp-1*, were recently reported to represent a class of epidermal progenitor cells, giving rise to *prog-1*[+], *AGAT-1*[+] mesenchymal cells, and other markers of epidermal cells (*van Wolfswinkel et al., 2014*). However, their precise molecular relationships and the mechanisms underlying the differentiation of these progeny cells are poorly understood.

Our results extend upon these studies by implementing a whole worm RNA-seq approach to identify genes expressed in *AGAT-1*[+] cells and their progeny and to characterize the post-mitotic transition states downstream of epidermal progenitors. We utilized a combination of assays, including whole-mount FISH, spatiotemporal kinetics after irradiation, and BrdU lineage tracing, to establish the temporal order of epidermal cell differentiation. Combinatorial FISH revealed that *zpuf-6* is co-expressed at varying levels in virtually all *AGAT-1*[+] cells, additional *AGAT-1*[neg] mesenchymal cells (*AGAT-1*[neg]*zpuf-6*[+]), and cells evenly distributed throughout the dorsal and ventral epidermis (*Figure 2*). We examined the expression patterns of all other genes from our screen that displayed discrete mesenchymal cell patterns in combination with *AGAT-1* and *zpuf-6*, and found that they all overlapped extensively. Many of these genes, such as *pla2* and *odc-1*, exhibited moderate

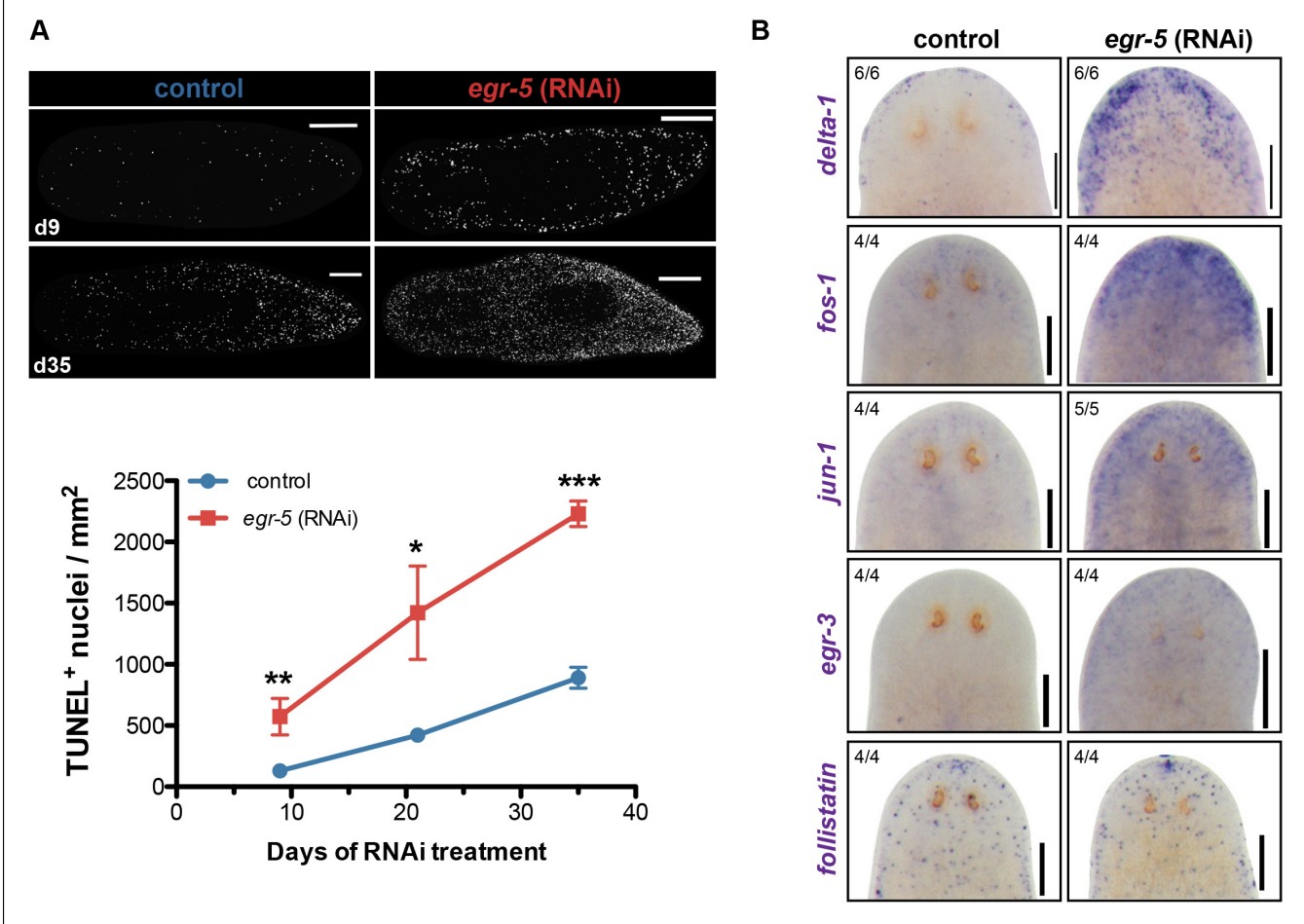

**Figure 7.** A global stress response is induced by loss of *egr-5*. (**A**) Whole-mount TUNEL assay measuring apoptosis in control and *egr-5*(RNAi) animals (8 feedings). Quantification of TUNEL-positive nuclei per surface area is plotted. Data represent means; error bars: SEM. Student's t test: *, p <.05, **, p <.005, ***, p <.0001. Representative maximum intensity projection of TUNEL-stained image for day 9 and day 35 are shown. Scale bars: 200 μm. (**B**) Wound-induced genes are up-regulated in *egr-5*(RNAi) animals. Representative colorimetric WISH images of *delta-1, fos-1, jun-1 egr-3 and follistatin* expression in the anterior regions of intact control and *egr-5*(RNAi) animals at 4Fd21 of RNAi treatment. Scale bars: 200 μm.

The following figure supplements are available for Figure 7:

**Figure supplement 1.** Wound-induced genes are upregulated in *chd4* and *p53* RNAi datasets.

expression in $AGAT\text{-}1^{neg}$ $zpuf\text{-}6^+$ cells, but expression was noticeably reduced in $zpuf\text{-}6^+$ epidermal cells (***Figure 2—figure supplement 3***), suggesting that these cells are turning off expression of those genes. In addition, *AGAT-1* and *zpuf-6* showed conserved spatiotemporal down-regulated kinetics after irradiation and in RNAi conditions leading to loss of $AGAT\text{-}1^+$ cells (***Figure 3***, ***Figure 3—figure supplement 1,2***), and a pulse of BrdU incorporated into dividing neoblasts showed a temporal progression through $prog\text{-}1^+$, $AGAT\text{-}1^+$ and $zpuf\text{-}6^+$ cells (***Figure 4A***). Altogether, we interpret these observations to mean that *prog-1, AGAT-1* and *zpuf-6* are markers of three major spatially and temporally related cell populations representing distinct transitional stages of epidermal maturation (***Figure 8***).

Differentiation and migration toward the mature planarian epidermis are temporally correlated, but what are the fates of cells once they pierce through the basement membrane and integrate into the epidermal layer? We propose that $zpuf\text{-}6^+$ epidermal cells may represent a stable transition state with the potential to give rise to an unknown number of differentiated cell types (***Figure 8***). $Zpuf\text{-}6^+$ epidermal cells are interspersed throughout the epidermis but do not express high levels of a cilia gene, *rootletin* (***Figure 4C***), which is a marker for differentiated ciliated epidermal cells.

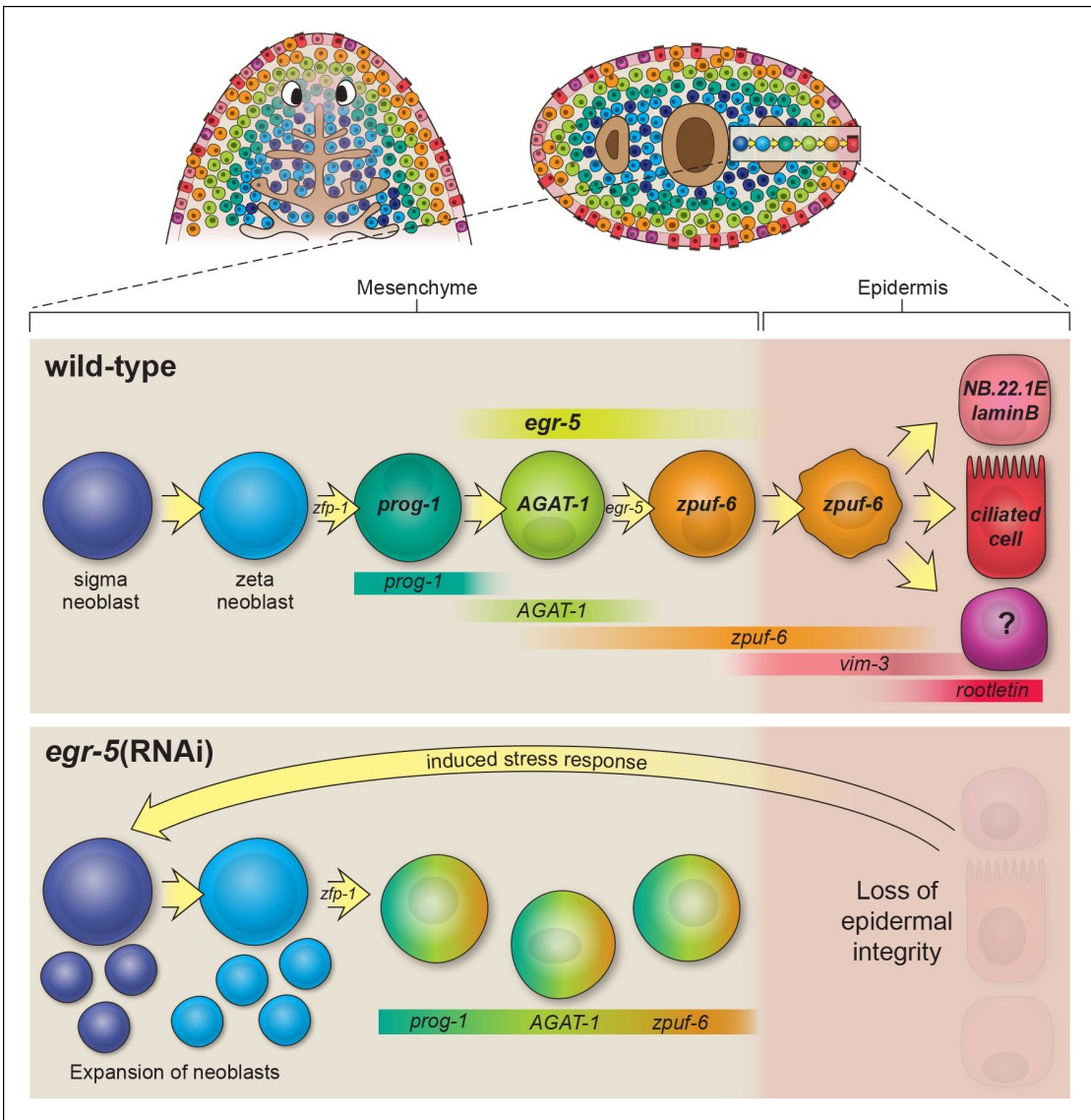

**Figure 8.** Model of planarian epidermal lineage progression. Schematic representation of the distribution of neoblasts, epidermal progeny cells in the mesenchyme, and differentiated epidermal cells along the anteroposterior (top left, dorsal view) and dorsoventral (top right, cross section) axes. Gut branches are shown as reference. Below: sigma-class neoblasts give rise to zeta-class neoblasts, which are epidermal progenitors. *zfp-1* is required to generate epidermal progeny cells, which begin to express different markers as cells undergo multiple transitions and intercalate into the mature epidermis. Gradient boxes represent domains and relative expression intensity of specified genes. *egr-5* is most strongly expressed in *AGAT-1*$^+$ cells and is required for proper differentiation of epidermal post-mitotic progeny cells. *zpuf-6*$^+$ epidermal cells likely represent a branching point in the epidermal cell fate decision and can generate multiple different cell types including marginal adhesive cells (*NB.22.1E* and *laminB*), multiciliated cells (*rootletin*), and other unknown cell types. Bottom panel: schematic of *egr-5*(RNAi) phenotype. Loss of *egr-5* disrupts the proper spatiotemporal transition of post-mitotic epidermal cells, resulting in *prog-1, AGAT-1* and *zpuf-6* to be co-expressed in the same cell, and leading to a depletion of mature epidermal cells. The resulting loss of epidermal integrity is sensed by the neoblasts (induced stress response) and causes an expansion of neoblasts and multiple progenitors before animals eventually lyse due to irreparable loss of an intact epidermal barrier.

However, *zpuf-6*$^+$ epidermal cells overlap extensively with a predicted intermediate filament (IF) gene, *vim-3*, which is expressed more broadly in the dorsal epidermis. Zpuf-6$^{neg}$ epidermal cells express lower levels of *vim-3* than *zpuf-6*$^+$ cells, but these cells begin in turn to express *rootletin*

(*Figure 4C*). IF proteins are a diverse class of cytoskeletal elements that provide cells with mechanical support, contribute to the structural integrity of cells, and act as markers of morphogenesis and differentiation (*Kim and Coulombe, 2007*). In addition, *zpuf-6*[+] cells overlap with two specific subset of cells: one expressing the marker *NB.22.1E* surrounding the ventral mouth opening and the other expressing the *laminB* IF gene that marks adhesive cells along the animal body margin (*Figure 4—figure supplement 3*). Lastly, *zpuf-6*[+] cells are present at higher cell density in the regenerating blastema than under normal homeostasis conditions (*Figure 4—figure supplement 2*), suggesting that they have not completed differentiation.

The decisions influencing a particular cell fate path for *zpuf-6*[+] epidermal cells could be determined by the immediate, surrounding environment. Given that dorsal and ventral epidermal cells are morphologically distinct, positional cues and signals are likely to contribute to the cell fate specification process. For example, transplantation experiments involving a plug of tissue from a donor transplanted in the reverse dorsal-ventral (DV) orientation with regard to the host can trigger tissue outgrowths on both sides where the grafted tissue maintains its original DV identity (*Kato et al., 1999*). The sub-epidermal layer of body wall muscle cells express most of the polarity cues that organize the planarian body axes, including receptors, ligands and inhibitors of the BMP and Wnt signaling pathways (*Witchley et al., 2013*), and could potentially impact epidermal fates as well. The role of Notch signaling has not been well characterized in planarians but has been implicated in the regulation of cell fate determination in epithelial cells of the mouse intestine (*VanDussen et al., 2012*) and the lung (*Rock et al., 2011*; *Rock and Hogan, 2011*). We cannot exclude the possibility that *zpuf-6*[+] epidermal cells simply reflect an abundant, differentiated cell population that undergoes rapid turnover. Additionally, there could be other progenitors (zeta-class dependent or independent) that give rise to specific epidermal cell types that have not been defined molecularly. To build upon our understanding of epidermal cell fate specification, it will be necessary to define all the types and numbers of cells that ultimately compose both the ventral and dorsal planarian epidermis, as well as the factors that contribute to their identity and maintenance.

## *Egr-5* is required for post-mitotic epidermal lineage progression

In planarians, TFs play prominent roles in specifying neoblasts to a committed fate (*Reddien, 2013*). Key TFs that are important for specifying neoblasts during homeostasis and regeneration often remain expressed in mature cell types, suggesting that these TFs are also required for the maintenance of cellular identity. Studies of planarian eye regeneration currently provide the most detailed characterization of planarian lineage specification. During regeneration, eye progenitors express conserved TFs *ovo, six-1/2* and *eya* at a distance from the eye primordium, forming a trail of cells and undergoing changes in gene expression as they migrate toward the primordium, where differentiation markers begin to be expressed (*Lapan and Reddien, 2011*; *Lapan and Reddien, 2012*). However, *ovo, six-1/2* and *eya* are also expressed in mature eyes and are likely involved in the maintenance of the optic cup cells and photo-sensing neurons of this sensory organ.

Our findings also suggest that distinct molecular changes occur both spatially and temporally, and are necessary for the proper differentiation of epidermal progenitors into mature cell types. However, *egr-5,* which is required for the stable maturation of epidermal progenitors and their progeny, does not appear to be required for the maintenance of the differentiated state. In fact, the lack of *egr-5* expression in the epidermis suggests that shutting off its expression may be an important step in promoting differentiation. Whether this regulation is intrinsic or extrinsic awaits further investigation. Moreover, because loss of *egr-5* leads to the misexpression of *AGAT-1* and *zpuf-6* markers in *prog-1*[+] cells, it could be directly or indirectly involved in turning off *prog-1* expression and activating the expression of genes required for the *AGAT-1*[+] transition stage. A key remaining question is how individual epidermal progeny cells undergo distinct differentiation transition stages, including responses to external signals or potential changes in chromatin structure.

## A physiological role for *AGAT-1*[+] progeny cells

Genes involved in small molecule metabolic processes (specifically organonitrogen processes) were enriched in our common *chd4* and *p53* RNAi down-regulated gene list. Many of the metabolic genes screened by in situ were expressed in *AGAT-1*[+] and *zpuf-6*[+] cells, including *gatm, odc-1*, cytochrome p450s (*cyp*), and *pla2*. Furthermore, genes including *acsl-2, adss, slc25a-19* and *zpuf-7* were

expressed in both the gut and cells of epidermal lineage, which suggests that these tissues may share common functional roles.

There are four arginine:glycine amidinotransferases (*AGAT-1, AGAT-2, AGAT-3, gatm*) identified in planarians. These enzymes catalyze the first step in creatine biosynthesis by transferring the amidino group of arginine to glycine to yield ornithine and guanidinoacetic acid (*Wyss, 2000*). Creatine plays an important role in muscle energy homeostasis. In vertebrates, de novo synthesis of creatine mainly takes place in the kidney, pancreas and liver (*Nabuurs et al., 2013*). Thus, it is assumed that *AGAT-1$^+$* cells synthesize creatine, which is then released and taken up by neighboring muscle cells and neurons (*Eisenhoffer et al., 2008*), though this has not been formally demonstrated.

Polyamines are ubiquitous polycations that are essential for eukaryotic cell growth, and polyamine metabolism is frequently dysregulated in cancer (*Pegg, 2009*). Some of the elucidated roles for polyamines in cell growth include maintenance of chromatin conformation, gene regulation, ion channel regulation, and free radical scavenging (*Casero and Marton, 2007*). Ornithine decarboxylase (ODC) catalyzes the first rate-limiting step in polyamine biosynthesis by converting ornithine to putrescine, which then is converted to spermidine and spermine. In planarians, ODC activity has been reported to be induced early in regeneration near the wound site, and inhibition of ODC activity appears to cause a reduction in cell proliferation and differentiation (*Saló and Baguna, 1989*). However, mechanisms of action by *odc-1* and the regulation of neoblasts require further characterization.

The cytochrome P450s (CYPs) are enzymes that use molecular oxygen and complex reaction chemistry to modify their substrates. They are involved in a large number of physiological processes, including steroid hormone synthesis and detoxification of xenobiotics (*Anzenbacher and Anzenbacherova, 2001*; *Zhang and Yang, 2009*). CYPs make up one of the most diverse eukaryotic gene families, and are classified into clans, families and sub-families based on phylogenetics and sequence identity (*Nelson et al., 2013*). We identified a number of *cyp* genes that were expressed in *AGAT-1$^+$* progeny cells as well as in the gut. Steroid metabolism and detoxification are two processes generally performed by the liver, raising the questions of what physiological roles these various planarian *cyp* genes play and whether *AGAT-1$^+$* cells may have an additional endocrine-like function.

Although p53 is best known for its central role as a tumor suppressor, it also regulates several aspects of cellular metabolism, including autophagy, central carbon metabolism, and lipid metabolism (*Berkers et al., 2013*). Fatty acids (FA) have essential roles in the cell as sources of energy, membrane components, and signaling molecules (*Lopes-Marques et al., 2013*). Long-chain fatty-acid coA-ligases (ACSL) are key enzymes involved in the initial steps of FA metabolism. We identified three enzymes involved in fatty acid metabolism, *acsl-1, acsl-2* and *pla2,* that are expressed in the gut as well as in epidermal progeny cells and the mature epidermis. Taken together, the presence of multiple metabolic genes co-expressed in *AGAT-1$^+$* cells and their progeny suggests that despite these cells being in a temporal stage transitioning to mature epidermal cells, they may also play a dual functional/physiological role in planarian homeostasis.

## Epidermal integrity and feedback regulation

Epithelia are a hallmark of multicellular organisms, forming the interface between the organism and the external environment and lining the cavities and surfaces of organs (*Donati and Watt, 2015*). The planarian epidermis provides the first response to amputation-induced injury by covering up the wound site within 30 min (*Reddien and Sánchez Alvarado, 2004*). It has been postulated that rhabdites are released at wound sites, where their contents produce a protective mucosal covering, possibly providing immunological functions (*Reisinger and Kelbetz, 1964*). Therefore, a breach in epidermal integrity could be the stimulus for initiating regeneration, though how this signal could be transduced to neoblasts is unknown.

The *egr-5*(RNAi) phenotype resulting in the expansion of neoblasts and multiple progenitor populations reveals that complex dynamics are at play between differentiating progeny and how neoblasts respond to loss of tissue integrity. It will be interesting to investigate the neoblast response in other RNAi or pharmacological contexts that directly affect epidermal integrity to determine how these perturbations affect neoblast function. However, it still remains possible that neoblast expansion could be a response to the global increase in apoptosis, although we still do not understand how the processes of cell proliferation and cell death are integrated. Alternatively, *egr-5* may be responsible for the expression of a signaling molecule to which neoblasts respond in a feedback control process. Feedback signaling from differentiating progeny to neoblasts may be an important and

robust strategy to control stem cell activity during homeostasis and regeneration. Therefore, unveiling the downstream targets of *egr-5* will help discern these possibilities, as well as provide more mechanistic insight on the role of *egr-5* and the progression of epidermal fate maturation.

Several key questions emerge from our findings. Why does epidermal lineage progression require multiple transition states? Do neoblast progeny contribute to the dynamics of the microenvironment, thereby playing a niche-like role? How are cell autonomous and non-cell autonomous mechanisms coupled for the maintenance of cellular equilibrium at the organismal level? How is the overall physiological status of the animal sensed, integrated and converted to the appropriate neoblast output? With a more nuanced understanding of their adaptive behavior, planarian neoblasts and the epidermal lineage have emerged as an important model system in which to study the dynamics between stem cell self-renewal and the orchestration of progeny differentiation.

## Materials and methods

### Planarian maintenance and irradiation exposure

*Schmidtea mediterranea* CIW4 asexual strain was maintained in 1X Montjuic salts supplemented with 50 µg/ml Gentamicin and fed homogenized calf liver paste as previously described (*Gurley et al., 2008*; *Reddien et al., 2005a*). Animals were on average starved between 7–10 days prior to starting experiments and ranged in size from 2 mm to 8 mm. Animals were exposed to 6000 Rads of gamma irradiation using a GammaCell 40 Exactor irradiator.

### Gene cloning and RNAi feeding experiments

Genes in this study were cloned from a CIW4 cDNA library into pPR-T4P vector (J. Rink) as described elsewhere (*Adler et al., 2014*). Primer sequences are provided in *Supplementary file 4*. Cloned gene vectors were transformed into bacterial strain HT115 for dsRNA production. RNAi food was prepared by mixing 50 ml of pelleted culture with 250 µl of calf liver paste (2X) or 50 ml of culture with 125 µl of calf liver paste (4X). Animals were fed every 3 days, with the first day designated as Day 0 of RNAi treatment. The number of RNAi feedings performed for *egr-5* knockdown varied depending on starting-size of animals and RNAi food concentration. RNAi feedings and time points analyzed are noted in figure legends as XFdY (3Fd18 = 3 RNAi feedings, day18). For all RNAi feeding experiments, *unc22* dsRNA was used as the control for the same number of feedings as experimental RNAi animals. Images of live animals were captured using a Leica M205 stereoscope.

### Whole-mount immunohistochemistry, in situ hybridizations and TUNEL staining

Whole-mount colorimetric and fluorescent in situ hybridizations were performed using a detailed protocol as previously described (*King and Newmark, 2013*; *Pearson et al., 2009*). Fluorescence-labeled animals were mounted in ScaleA2 solution (*Hama et al., 2011*). Immunostaining with anti-H3P (1:1000, Millipore, Billerica, MA) was performed following fluorescent in situ development and was detected using Alexa-conjugated secondary antibody (1:1000, Abcam, Cambridge, MA). Animals were fixed and stained for TUNEL using a method previously described (*Pellettieri et al., 2010*) with modifications: animals were bleached in 0.075% ammonia and 3% hydrogen peroxide and treated with ProteinaseK (2 µg/ml) in PBSTx (0.3% Triton) for 10 min followed by 4% formaldehyde incubation for 10 min prior to TdT reaction.

### Histology and SEM

Cryosectioned animals were processed after fluorescent in situ development as previously described (*Tu et al., 2012*). For nuclear fast red staining, animals were processed for colorimetric ISH and were subsequently fixed overnight in 4% paraformaldehyde (PBS) at 4°C followed by dehydration in 30, 50, and 70% ethanol. Fixed specimens were embedded in paraffin and serial sectioned at 10 µm thickness and counter stained with nuclear fast red.

For scanning electron microscopy, animals were immersed in a relaxant fixative (1% HNO$_3$, 0.85% formaldehyde, 50 mM MgSO$_4$) as described elsewhere (*Rompolas et al., 2013*) for 5 min and were replaced with fresh fixative and rocked overnight at room temperature. Animals were then transferred to a solution containing 2.5% glutaraldehyde, 2% paraformaldehyde, 1% sucrose, 1 mM

CaCl2 in 0.05 M NaCacodylate buffer pH 7.36 and left at 4°C until ready for processing. Animals were rinsed in ultrapure water, and secondary fixation was performed at 4°C overnight in 2% aqueous osmium tetroxide. Samples were dehydrated in a graded series of ethanol and dried in a Tousimis Samdri-795 critical point dryer. Samples were mounted on stubs and sputter coated with gold palladium. Imaging was done with a Hitachi TM-1000 tabletop SEM.

## Image analysis and quantifications

Colorimetric WISH images were acquired using a Zeiss Lumar V12 stereomicroscope equipped with an AxioCam HRc. Colorimetric WISH tissue sections were imaged using a Zeiss Axiovert206. Fluorescent images were acquired with either a Zeiss LSM-510 VIS or a Perkin Elmer Ultraview VOX spinning disk. Stitching and batch processing of images, and H3P quantifications per surface area were performed as previously reported (*Adler et al., 2014*). For H3P/progenitor colocalization thresholding, spots were selected based on signal intensity a multiple above the noise background and filtered based on size. For fluorescence intensity quantification, Z-stack images were acquired and z-projected by average in a fixed neighborhood around the bottom of the animal as determined by DAPI nuclear staining after background subtraction. For epidermal cell quantifications, single focal plane images were acquired and nuclei within a fixed sized rectangle in regions of uniform cell density were quantified using 'Find Maxima' with Fiji software. All macros and plugins are available at https://github.com/jouyun.

## BrdU labeling and quantification

BrdU was administered by soaking animals for 24 hr in 20 mg/ml BrdU (Sigma, St. Louis, MO) dissolved in 3% DMSO and 1X Montjuic. Animals were then incubated in 5 g/L Instant Ocean supplemented with 50 µg/ml Gentamicin over the course of BrdU chase period. Animals were fixed and processed using the in situ hybridization protocol except bleached in 6% $H_2O_2$ in PBSTx (0.3% Triton) for 3–4 hr under direct light. After development, specimens were treated with 2N HCl for 45 min at room temperature. BrdU was detected using a rat anti-BrdU antibody (1:1000; Abcam, Cat. No. ab6326). Primary antibody was detected with HRP-conjugated anti-rat antibody (1:1000; Jackson ImmunoResearch, West Grove, PA). Images were acquired with a Perkin Elmer (Waltham, MA) Ultraview VOX spinning disk with a 20x 0.8 NA Plan Apochromat objective (Zeiss, Oberkochen, Germany) onto an Orca R2 (Hamamatsu Photonics, Hamamatsu, Japan) camera. Integration times were adjusted per worm to achieve well-saturated images. Z-stacks were collected at the anterior end of the worm over 250 µm in the Z dimension, with 2 µm steps, which contained nearly the entire depth of the worm. Images were cropped to a 400 $pixel^2$ (132 $µm^2$) region posterior to the brain. The stack was also cropped in Z to the ventral half of the stack. Target-specific cells were first marked manually in Fiji followed by target and BrdU double-positive cells. At least 6 worms (up to 10) per time point were analyzed and roughly 6,500 cells total were quantified. The average percent positive cells were reported ((double-positive/total)*100). Representative images (*Figure 4A*) were first background subtracted in ImageJ with a 25 pixel radius rolling ball and then gaussian-blurred with a 1 pixel radius.

## RNA-seq and GO enrichment analyses

For RNA-seq analysis of control, *chd4*(RNAi) and *p53*(RNAi) animals, three biological replicates of 10 worms each were collected for RNA isolation. mRNAseq libraries were generated from 500ng of high quality total RNA, as assessed using the LabChip GX (Perkin Elmer). Libraries were made according to the manufacturer's directions for the TruSeq Stranded mRNA LT– set A and B (Illumina, San Diego, CA; Cat. No. RS-122-2101 and RS-122-2102). Resulting short fragment libraries were checked for quality and quantity using a LabChip GX and Qubit Fluorometer (Life Technologies, Carlsbad, CA). Libraries were pooled, requantified and sequenced as 50 bp single reads on the Illumina HiSeq 2500 instrument using HiSeq Control Software 2.0.5. Following sequencing, Illumina Primary Analysis version RTA 1.17.20 and Secondary Analysis version CASAVA-1.8.2 were run to demultiplex reads for all libraries and generate FASTQ files. RNA-seq analysis was carried out by mapping sequence reads to a set of 36,035 *S. mediterranea* transcripts assembled from various sources including a previous transcriptome used in microarray studies (*Adler et al., 2014*), trinity assemblies from lab-generated data involving whole animals, embryos, and sorted X1 cells, a

transcriptome from the Bartscherer lab (*Boser et al., 2013*), and the Dresden transcriptome assembly from PlanMine (http://planmine.mpi-cbg.de), reduced as a collection to unique representations of loci via CD-HIT (*Fu et al., 2012*). Sequences can be downloaded from http://smedgd.stowers.org. Reads were aligned using bowtie with the following parameters: –best –strata -v 2 -m 5, and read counts to genes were tallied from the SAM files with a custom script. Differential gene expression was evaluated using R and the edgeR library (*Robinson et al., 2010*). P-values were adjusted as previously described (*Benjamini and Hochberg, 1995*). The RNA-Seq data and the transcriptome against which it was quantified have been archived at GEO and is available under accession number: GSE72389. Gene Ontology (GO) (*Gene Ontology Consortium, 2015*) terms were assigned to each *S. mediterranea* gene based on homologous PFAM domains and significant Swissprot hits. GO term enrichment was performed using the R package topGO (*Alexa and Rahnenfuhrer, 2010*).

## Acknowledgements

We would like to thank all members of the Sánchez Alvarado lab for their support, especially Carrie Adler and Erin Davies for helpful comments on the manuscript. We thank Beth Duncan for kindly sharing X1 data and Sarah Elliott and Aurimas Gumbrys for RNA probes. We thank Eric Ross and Kirsten Gotting for bioinformatics assistance, Jay Unruh for statistical assistance, David Nelson (University of Tennessee Health Science Center) for planarian *CYP* nomenclature, and Mark Miller for illustration assistance. We also acknowledge Melania McClain for SEM assistance, Allison Peak for RNA-seq library assistance and all other members of the Histology, Microscopy, Molecular Biology and Planaria core facilities at the Stowers Institute for their technical support. This work was supported by NIH R37GM057260 to ASA. KCT was an Ellison Medical Foundation/AFAR Fellow of the Life Sciences Research Foundation. ASA is an investigator of the Howard Hughes Medical Institute.

## Additional information

### Competing interests

ASA: Reviewing editor, *eLife*. The other authors declare that no competing interests exist.

### Funding

| Funder | Grant reference number | Author |
| --- | --- | --- |
| National Institute of General Medical Sciences | R37GM057260 | Alejandro Sánchez Alvarado |
| Howard Hughes Medical Institute | | Alejandro Sánchez Alvarado |
| Stowers Institute for Medical Research | | Alejandro Sánchez Alvarado |
| Ellison Medical Foundation | | Kimberly C Tu |
| American Foundation for Aging Research | | Kimberly C Tu |

The funders had no role in study design, data collection and interpretation, or the decision to submit the work for publication.

### Author contributions

KCT, Conception and design, Acquisition of data, Analysis and interpretation of data, Drafting or revising the article; LCC, JJL, SAMcK, CWS, Acquisition of data, Analysis and interpretation of data, Drafting or revising the article; HTKV, Analysis and interpretation of data, Drafting or revising the article, Contributed unpublished essential data or reagents; ASA, Conception and design, Analysis and interpretation of data, Drafting or revising the article

### Author ORCIDs

Alejandro Sánchez Alvarado, (iD) http://orcid.org/0000-0002-1966-6959

## Additional files

• Supplementary file 1. Summary of down-regulated genes in *chd4* and *p53* RNAi RNA-seq data sets. Tab1, down-regulated genes in *chd4*(RNAi), *p53*(RNAi), and the 587 common down-regulated genes in both *chd4* and *p53* RNAi data sets. Tab 2, Top BLAST hits of 587 common down-regulated gene list to *C. elegans, D. melanogaster, M. musculus*, and *H. sapiens* NCBI Refseq with an e-value cutoff of .001. Tab 3, GO term enrichment analysis for the common down-regulated gene list.

• Supplementary file 2. Summary of up-regulated genes in *chd4* (d15) and *p53* (d18) RNAi RNA-seq data sets. Genes were classified as up-regulated if the adjusted p value <1e-13 and log2 fold change >1.

• Supplementary file 3. In situ expression summary of cloned genes from *chd4* and *p53* RNAi data sets. Tab 1, List of 147 genes cloned and screened by colorimetric WISH. Tab 2, Summary of 29 candidate genes chosen from in situ screen (*Figure 1E*, *Figure 2*) characterized by triple FISH with *AGAT-1* and *zpuf-6*.

• Supplementary file 4. Primer sequences of genes cloned in this study.

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
