## [Decision Letter]

Thank you for submitting your work entitled "The planarian epidermis as a model for adult stem cell specification and differentiation" for peer review at *eLife*. Your submission has been favorably evaluated by Fiona Watt (Senior editor) and three reviewers, one of whom, Yukiko Yamashita, is a member of our Board of Reviewing Editors.

The reviewers have discussed the reviews with one another and the Reviewing editor has drafted this decision to help you prepare a revised submission.

The study addresses an important topic of the molecular mechanisms controlling fate transitions among post-mitotic cells. This paper makes a significant contribution to our understanding of epidermal differentiation in planarians. It identifies many useful new markers for characterizing this process, in particular *egr-5* as a gene required for maturation of post-mitotic progenitors forming epidermis. The paper is well written, and the data are presented beautifully.

Essential revisions:

1) As the authors point out, other prior work made significant contributions to understanding epidermal differentiation in planarians, and so it is an overstatement to claim (in the Abstract, title, and other locations in the text) that this study establishes the planarian epidermis as a model for studying differentiation. The title in particular should be changed to reflect the precise major contributions this study makes well (implication of *egr-5* in control of post-mitotic cell maturation in epidermal differentiation).

2) Need of additional evidence in support of the model that epidermal defects trigger a wound response. At the moment, this great idea is based on the expression of a single marker, *delta-1* a known wound-responsive gene. Because the role of *delta-1* in the planarian epidermis has not been reported, it is possible that *delta* could also be activated based on a role it plays in epidermal differentiation. The authors should strengthen this hypothesis by qPCR experiments (or other means) to measure whether other wound-response genes are also upregulated after *egr-5* RNAi. Are wound-response genes upregulated in their *chd4* and *p53* (RNAi) datasets? As there is no functional evidence showing that a loss of epidermal integrity in *egr5* RNAi leads to global proliferation/apoptosis, in parallel to conducting some of the critical and feasible experiments. We also recommend toning down the language, if necessary, depending on how much the claim can be strengthened with additional experiments that are feasible and within the scope of revision.

3) Lineage hierarchy described in Figure 8 is not entirely supported by the experimental data. There are a few straightforward experiments that can be done to strengthen this model. Also, the authors could soften the conclusion where the straightforward experiments are not possible. Several examples of possible experiments: Does *zfp-1* RNAi reduce *zpuf-6* and *vim-3* expression? How does the *zfp-1* RNAi RNA-seq dataset compare with the *p53* RNAi and *chd4* RNAi datasets (were *egr-5* and *zpuf-6* previously identified in that dataset)? What is the phenotype of *zpuf-6* RNAi animals? Do *egr-5* RNAi animals lose or fail to form *nb22.12e/laminB*-expressing cells?

---

## [Author Response]

*1) As the authors point out, other prior work made significant contributions to understanding epidermal differentiation in planarians, and so it is an overstatement to claim (in the Abstract, title, and other locations in the text) that this study establishes the planarian epidermis as a model for studying differentiation. The title in particular should be changed to reflect the precise major contributions this study makes well (implication of* egr-5 *in control of post-mitotic cell maturation in epidermal differentiation).*

We thank the reviewers for pointing this out. Accordingly, we have changed the title of the manuscript to reflect the specific findings reported in the manuscript. It now reads: “*Egr-5* is a post-mitotic regulator of planarian epidermal differentiation.” We have also modified the wording in the Abstract and other locations in the text to indicate that our results further support, the utility of the planarian epidermis as a model for the molecular study of post-mitotic differentiation.

*2) Need of additional evidence in support of the model that epidermal defects trigger a wound response. At the moment, this great idea is based on the expression of a single marker,* delta-1 *a known wound-responsive gene. Because the role of* delta-1 *in the planarian epidermis has not been reported, it is possible that* delta *could also be activated based on a role it plays in epidermal differentiation. The authors should strengthen this hypothesis by qPCR experiments (or other means) to measure whether other wound-response genes are also upregulated after* egr-5 *RNAi. Are wound-response genes upregulated in their* chd4 *and* p53 *(RNAi) datasets? As there is no functional evidence showing that a loss of epidermal integrity in* egr5 *RNAi leads to global proliferation/apoptosis, in parallel to conducting some of the critical and feasible experiments. We also recommend toning down the language, if necessary, depending on how much the claim can be strengthened with additional experiments that are feasible and within the scope of revision.*

We agree with the reviewers that the induced expression of *delta-1* alone may be insufficient evidence to claim that epidermal defects trigger a wound response. As suggested, we performed WISH to look at the expression of 3 additional wound-induced genes that have been previously reported, including *egr-3, fos-1* and *jun-1*, which are homologs of the “immediate early genes.” All of these genes exhibit increased expression in *egr-5*(RNAi) animals at day 21 of RNAi treatment compared to controls. However, *follistatin*, a gene that is specifically induced in response to the loss of tissue, does not show an increase in expression in *egr-5*(RNAi) animals, which is expected since no tissue has been removed. Accordingly, Figure 7 and the accompanying texts have been modified to include these additional wound-induced genes (showing day 21 only).

We also examined the same 5 wound-induced genes in our *chd4* and *p53* RNAi data sets and found that *egr-3, fos-1* and *jun-1* were all significantly up-regulated through multiple time points of RNAi treatment. We added a plot of log2 ratios of these data in Figure 7—figure supplement 1. We argue that the activation of multiple wound-induced genes in *egr-5, chd4* and *p53* RNAi conditions support our hypothesis that defects in epidermal differentiation and/or loss of epidermal progeny lead to a loss of epidermal integrity, which may trigger a global wound response. However, because we do not have functional data to support that defects in epidermal integrity specifically induce proliferation/apoptosis (we believe this is beyond the scope of this paper), we have also toned down the language of our conclusion (the last sentence of the Results).

*3) Lineage hierarchy described in Figure 8 is not entirely supported by the experimental data. There are a few straightforward experiments that can be done to strengthen this model. Also, the authors could soften the conclusion where the straightforward experiments are not possible. Several examples of possible experiments: Does* zfp-1 *RNAi reduce* zpuf-6 *and* vim-3 *expression? How does the* zfp-1 *RNAi RNA-seq dataset compare with the* p53 *RNAi and* chd4 *RNAi datasets (were* egr-5 *and* zpuf-6 *previously identified in that dataset)? What is the phenotype of* zpuf-6 *RNAi animals? Do* egr-5 *RNAi animals lose or fail to form* nb22.12e/laminB*-expressing cells?*

We thank the reviewers for suggesting these experiments in order to strengthen our lineage hierarchy model presented in Figure 8. Using whole-mount ISH, both *zpuf-6* and *vim-3* expression are reduced in *zfp-1*(RNAi) animals, and this is already shown in Figure 2—figure supplement 1 (bottom row). Both *zpuf-6* and *egr-5* were identified in the *zfp-1*(RNAi) RNA-seq datasets, which substantiates our whole-worm RNA-seq approach in identifying genes enriched in post-mitotic epidermal progeny.

There is no discernible phenotype for *zpuf-6*, but that may be due to functional redundancy because we identified multiple novel secreted proteins similar to *zpuf-6* that are also co-expressed in similar cell populations. We have yet to simultaneously knock down all of these novel genes to identify their putative functions, as it is something we are interested in pursuing but will also require the development of additional assays. Momentarily, *zpuf-6* acts a useful marker that highlights an abundant epidermal post-mitotic cell population spanning both the mesenchyme and the epidermis.

We agree that evidence demonstrating the failure to form *NB.22.1E/lamin*B cells in *egr-5*(RNAi) animals would strengthen our lineage hierarchy model. Therefore, we quantified the number of *laminB+* cells in the anterior region because we show in Figure 4—figure supplement 2 (submitted version) that these anterior cells turnover more rapidly than cells in the posterior. We found that there is a significant reduction of *laminB*+ cells in *egr-5*(RNAi) animals compared to controls at 21d of RNAi treatment. These data are presented as an additional figure supplement (Figure 5—figure supplement 4) and the corresponding text has been added to the subsection “*egr-5* is a regulator of post-mitotic epidermal lineage fate specification”.